# Deforestation intensifies daily temperature variability in the northern extratropics

Jun Ge [1,2,7] ✉, Qi Liu [1,2,7], Beilei Zan[3,4,5], Zhiqiang Lin[6], Sha Lu[1,2], Bo Qiu [1,2] & Weidong Guo [1,2] ✉

While the biogeophysical effects of deforestation on average and extreme temperatures are broadly documented, how deforestation influences temperature variability remains largely unknown. To fill this knowledge gap, we investigate the biogeophysical effects of idealized deforestation on daily temperature variability at the global scale based on multiple earth system models and in situ observations. Here, we show that deforestation can intensify daily temperature variability (by up to 20%) in the northern extratropics, particularly in winter, leading to more frequent rapid extreme warming and cooling events. The higher temperature variability can be attributed to the enhanced near-surface horizontal temperature advection and simultaneously is partly offset by the lower variability in surface sensible heat flux. We also show responses of daily temperature variability to historical deforestation and future potential afforestation. This study reveals the overlooked effects of deforestation or afforestation on temperature variability and has implications for large-scale afforestation in northern extratropic countries.

Forests have undergone tremendous losses and gains due to human and natural disturbances in recent decades[1,2]. In addition to the impact on the carbon cycle, deforestation also has profound impacts on local and regional climate through biogeophysical processes[3]. On the one hand, deforestation enlarges the surface albedo and cools the climate; on the other hand, deforestation diminishes evapotranspiration and warms the climate due to the lower aerodynamic roughness, rooting depth, leaf area, and canopy conductance for transpiration[4–6]. The albedo-driven cooling effect dominates boreal regions, whereas the evapotranspiration-driven warming effect dominates the tropics[7–10]. In the mid-latitudes, however, the deforestation effect is complicated and uncertain because of the lower model agreement in this regard[11,12]. Deforestation also causes daytime warming and nighttime cooling effects over most regions of the world[9,13–15]. This diurnal asymmetry of temperature responses to deforestation amplifies the diurnal

temperature range[10,15]. Deforestation can further influence temperature extremes, particularly hot extremes[16–20]. On hot days, forests can maintain a relatively lower ambient temperature due to sustained evapotranspiration and more efficient heat dissipation[16]; thus, deforestation tends to aggravate hot extremes[16,19,20]. This effect, however, is uncertain because some studies also report alleviated hot extremes following deforestation[17,18].

Although the deforestation effects on the mean, the diurnal cycle, and extremes of temperature have been widely documented, how deforestation influences temperature variability remains largely unknown. Temperature variability describes fluctuations in a time series of temperature at various (from daily to decadal) time scales. In particular, the high-frequency daily temperature variability is closely related to human and natural systems. For example, enhanced daily temperature variability causes higher risks of mortality induced by

[1]School of Atmospheric Sciences, Nanjing University, Nanjing, China. [2]Joint International Research Laboratory of Atmospheric and Earth System Sciences, Nanjing University, Nanjing, China. [3]Collaborative Innovation Center on Forecast and Evaluation of Meteorological Disasters/Key Laboratory of Meteorological Disaster, Nanjing University of Information Science and Technology, Nanjing, China. [4]Ministry of Education/International Joint Research Laboratory on Climate and Environment Change, Nanjing University of Information Science and Technology, Nanjing, China. [5]School of Atmospheric Sciences, Nanjing University of Information Science and Technology, Nanjing, China. [6]School of Atmospheric Sciences, Chengdu University of Information Technology, Chengdu, China. [7]These authors contributed equally: Jun Ge, Qi Liu. ✉e-mail: junge@nju.edu.cn; guowd@nju.edu.cn

chronic disease[21–23] or epidemic disease[24,25]. Changes in daily temperature variability also influence ecosystem functions, such as crop yields[26–28] and coral bleaching[29]. Higher daily temperature variability can even threaten macroeconomic growth[30]. Statistically speaking, an increase in variability commonly coincides with a higher tail probability, implying more frequent weather and climate extreme events[31,32] and consequently widespread adverse impacts on nature and humans[33]. Therefore, understanding the deforestation effect on temperature variability is as important as understanding the effects on the temperature mean and extremes.

The historical daily temperature variability is observed to decrease overall in the northern mid- and high-latitude continents in boreal winter, spring, and autumn[34–38] but to increase in a few regions in boreal summer[37–39]. Changes in daily temperature variability were previously attributed to anthropogenic greenhouse gas (GHG) emissions[35–38], aerosols[36–38], urbanization[40], and internal climate variability[37,38] but were rarely linked to forest changes. Given the large-scale deforestation worldwide during the historical period[1,2], examining the deforestation effect on daily temperature variability can improve our understanding of human contributions to the evolution of daily temperature variability. Furthermore, afforestation has been broadly proposed as a nature-based solution to mitigate climate warming. This proposal is mostly built on the cooling effect of afforestation through carbon sequestration and some biogeophysical processes (e.g., the evaporative cooling effect)[41–43], but the afforestation effect on temperature variability has never been considered and evaluated. Therefore, understanding the afforestation effect on daily temperature variability can help to avoid unanticipated climatic consequences following the implementation of large-scale afforestation.

In this study, we investigate the biogeophysical effect of idealized deforestation on daily temperature variability at the global scale based on multiple earth system models and in situ observations. We also explore the potential mechanisms for the deforestation effect based on the thermodynamic energy equation. The results show that large-scale deforestation can intensify regional daily temperature variability in the northern extratropics, particularly in winter. As a result, the deforested areas suffer from more frequent rapid extreme warming and cooling events. The higher temperature variability is mainly attributed to the enhanced near-surface horizontal temperature advection (TADV) and simultaneously is partly offset by the lower variability in surface sensible heat flux (SHF). Moreover, we also detect higher daily temperature variability in North America due to large-scale deforestation since 1850. The deforestation effect even offsets the total effect of other anthropogenic forcings (e.g., GHGs) at regional scales. In contrast, large-scale afforestation in North America is projected to reduce daily temperature variability by the end of the twenty-first century. This study reveals the overlooked human influence on daily temperature variability through deforestation or afforestation, which has implications for policy decisions on the implementation of large-scale afforestation.

## Results
### The deforestation effect on daily temperature variability
We first examine the biogeophysical effect of deforestation on daily temperature variability on a global scale. To identify all hot-spot regions where daily temperature variability is susceptible to deforestation, the preindustrial control simulation (piControl) of the Coupled Model Intercomparison Project Phase 6[44] (CMIP6) and the idealized global deforestation simulation (deforest-globe) of the Land Use Model Intercomparison Project[45] (LUMIP) are used. For piControl, all external forcings are fixed at preindustrial levels (see Methods). deforest-globe is identical to piControl, except removing 20 million km² of forests globally in the first 50 years (Fig. 1a, b; see Methods); in the following 30 years, the total tree cover is maintained unchanged (Fig. 1b). Vegetation dynamics are required to be switched off in the

deforested grid cells for deforest-globe[45]. The biogeophysical effect of the idealized deforestation can be isolated through the comparison of the last 30-year (years 51–80) simulations of piControl and deforest-globe (deforest-globe minus piControl; see Methods). While such large-scale deforestation is unrealistic, using the idealized deforestation scenario still makes sense. First, this scenario enhances the intermodel consistency in the prescribed deforestation and thereby enables a more meaningful comparison of the simulated deforestation effects across the models[45]. Second, this scenario makes it easier to establish the deforestation signal from the model stochastic noise. Five earth system models are used here given the model output availability (Supplementary Table 1).

Daily temperature variability is quantified by the day-to-day variation index[34]. The day-to-day temperature variation (DTDT) is defined as the mean value of absolute differences in daily mean 2-meter temperature between every two neighboring days during a given time period (see Methods). We first evaluate the model performance in the DTDT simulation with two independent reanalysis datasets: the ERA5 reanalysis[46] and the NCEP-DOE AMIP-II reanalysis[47] (see Methods). Validated with the reanalysis, the models can reasonably simulate the spatial pattern of DTDT. DTDT is overall larger at higher-latitude continents (Supplementary Fig. 1a–d, i–p). Moreover, DTDT is larger in winter (December, January, and February), followed by spring (March, April, and May) and autumn (September, October, and November), and lowest in summer (June, July, and August) in the northern extratropics.

We also evaluate the model representation of the biogeophysical effect of deforestation on mean surface temperature with a satellite-based dataset[13,48]. This dataset provides the observed biogeophysical effect of deforestation at the global scale based on the "space-for-time" substitution method, that is, the comparison of the spatially adjacent forest and openland pixels[48] (see Methods). The models consistently suggest a cooling effect of deforestation in the northern extratropics (Supplementary Fig. 2a–d). In the tropics, however, the models show less agreement on the deforestation effect, and the multimodel mean suggests a warming effect of deforestation. The simulations are overall consistent with the observations (Supplementary Fig. 2e, f) and previous studies[7–10]. Note that, in contrast to the observations, the models simulate a more widespread cooling effect of deforestation in the northern mid-latitudes. This discrepancy is mainly attributed to the fact that the deforestation-induced atmospheric feedbacks (e.g., cloud formation) to surface temperature are fully considered in the models but absent in the observed signal due to the application of the "space-for-time" method[49,50]; this method assumes identical background climate over the paired forest and openland pixels. It has been demonstrated that the atmospheric feedbacks are considerable and mostly explain the simulated cooling effect of deforestation in the northern mid-latitudes[49,50].

Figure 1c–f shows the biogeophysical effect of deforestation on DTDT for each season. The multimodel mean result is shown here, and the black dots denote the model agreement on the sign of the DTDT change. The models suggest a widespread increase in DTDT in deforested areas in the northern extratropics, e.g., North America and Eurasia. The DTDT signal is stronger (up to 0.7 °C or 20%) in winter, moderate in spring and autumn (between 0.1 and 0.5 °C or 10 and 15%), and weaker in summer (between 0.05 and 0.3 °C or 5 and 10%). More importantly, all five models agree well on the sign of the DTDT change (see the black dots), implying a robust signal across the models. In other deforested areas (e.g., the tropics and the Southern Hemisphere), however, the DTDT response to deforestation is mostly negligibly small (within ± 0.05 °C or ±5%) and has low intermodel consistency.

Daily temperature variability can also be quantified by a more commonly used index, namely, the standard deviation of the daily mean 2-meter temperature during a given time period[32] (SDT; see Methods). The model behavior in the SDT simulation is also reasonable

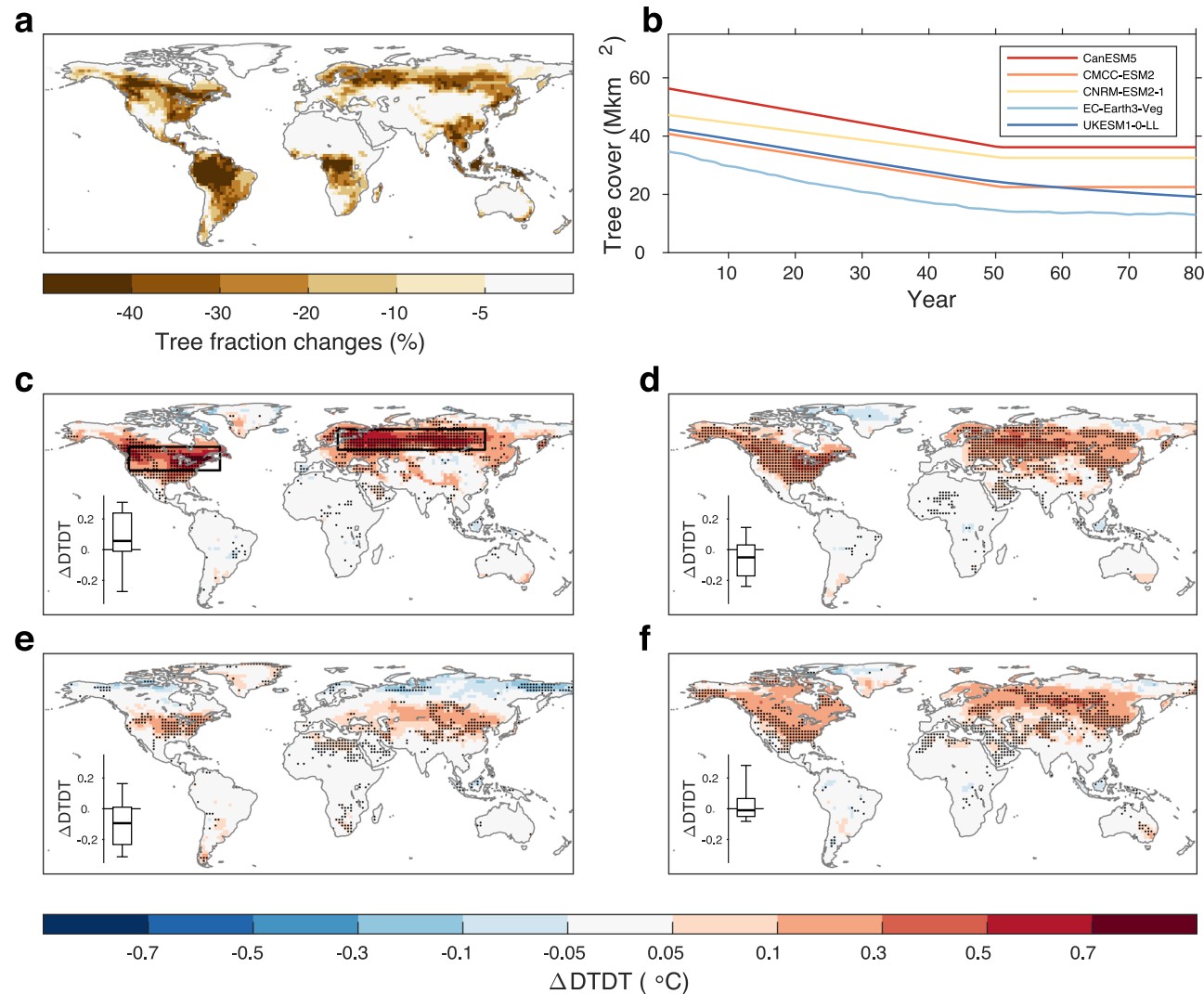

**Fig. 1 | Tree fraction changes in the idealized deforestation scenario and the biogeophysical effects on daily temperature variability. a** Multimodel mean tree fraction changes by the end of the deforest-globe simulation. **b** The evolution of tree cover for each model in deforest-globe. **c–f** The multimodel mean differences in day-to-day temperature variability (DTDT) in **c** DJF (December, January, and February), **d** MAM (March, April, and May), **e** JJA (June, July, and August), and **f** SON (September, October, and November) between the piControl and deforest-globe simulations (deforest-globle minus piControl). The black dots in **c–f** denote that all five models agree on the sign of the DTDT change. The black rectangles in **c** cover the two regions in North America (38–54°N, 60–122°W) and Eurasia (52–66°N, 20–120°E), which are further analyzed in Fig. 2. The box-and-whisker plots embedded in **c–f** show the DTDT differences between the paired forest and openland sites (openland minus forest; Supplementary Fig. 5) from the FLUXNET and AmeriFlux datasets. The horizontal black line of the box denotes the median. The bottom and top edges of the box denote the 25th and 75th percentiles, respectively. The whiskers extend to the 10th and 90th percentiles.

when validated against the reanalysis (Supplementary Fig. 3a–d, i–p). Similarly, the multimodel mean also suggests a widespread increase in SDT in the northern extratropical deforested areas, particularly in winter (Supplementary Fig. 4). The consistent DTDT and SDT responses to deforestation indicate that the result is independent of the metrics of daily temperature variability.

In addition to the simulations, we attempt to find observational evidence of the deforestation effect on daily temperature variability. To this end, 24 selected pairs of spatially adjacent forest and openland sites in North America and Europe (Supplementary Fig. 5 and Supplementary Table 3) from the FLUXNET and AmeriFlux datasets are used[51–53] (see Methods). The box-and-whisker plots imbedded in Fig. 1c–f show the DTDT differences between the paired forest and openland sites (openland minus forest) for each season. The DTDT differences across the paired sites are positively biased (with a mean value of 0.07 °C) in winter, implying overall higher daily temperature variability over openland sites than neighboring forest sites. This result confirms the enhanced wintertime daily temperature variability in the

northern extratropical deforested areas in the simulations. For other seasons, however, the DTDT differences across the paired sites are either negatively biased or unbiased. This result implies that the DTDTs of openland sites are overall smaller than or identical to the values of neighboring forest sites. This result is in contrast with the simulated result, which is discussed later.

**The deforestation effect on rapid warming and cooling events**

Since DTDT is calculated as the absolute value of temperature differences between neighboring days, DTDT itself only reflects the magnitude of daily temperature variability. In fact, the temperature difference between neighboring days ($\delta T$) can be either positive or negative, and positive and negative $\delta T$ values indicate rapid warming and cooling events, respectively. Thus, the following question emerges: how do rapid warming and cooling events respond to deforestation and contribute to the increased DTDT?

To answer this question, Fig. 2 shows the probability density distributions of $\delta T$ before and after deforestation for each season in the

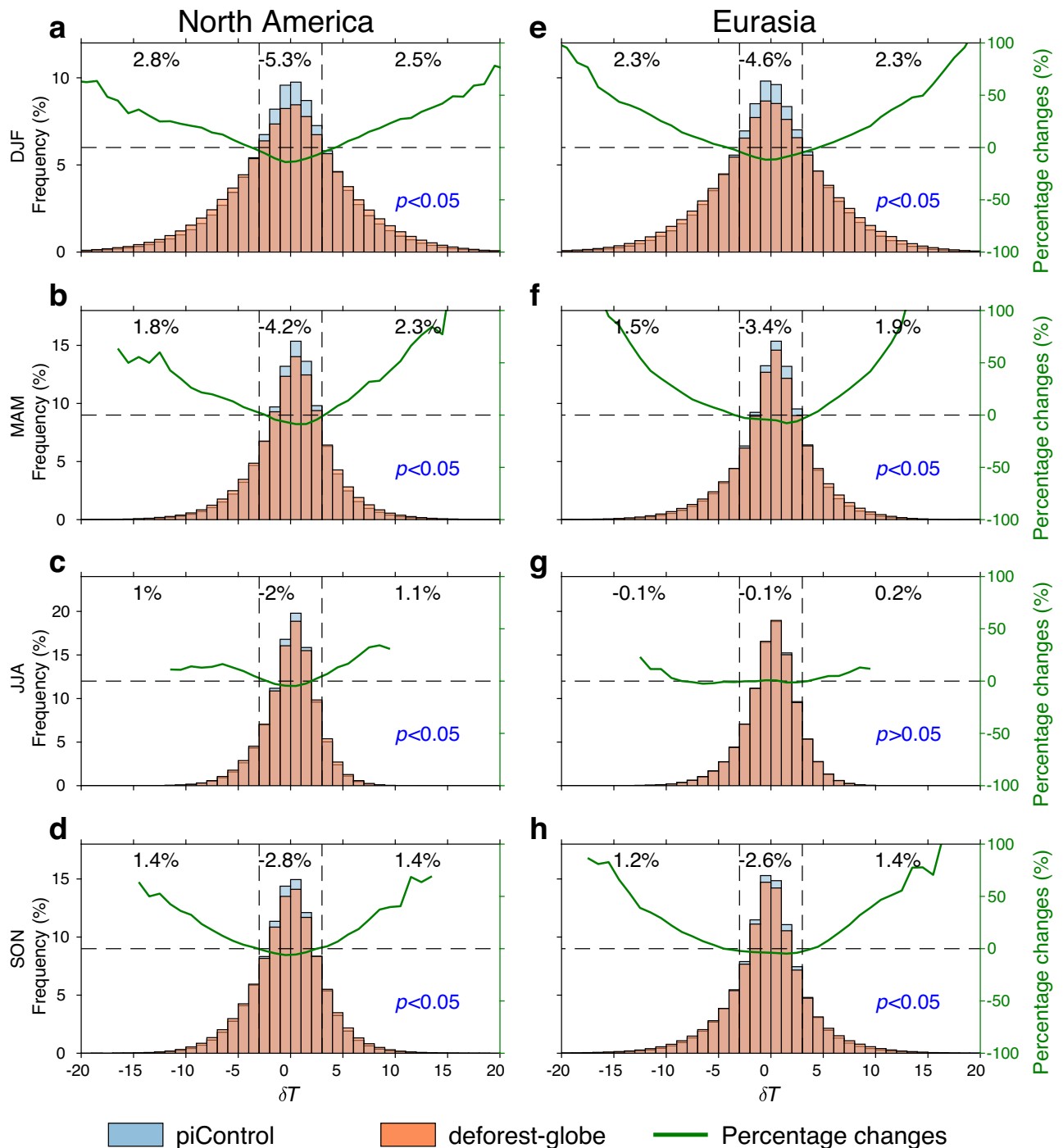

**Fig. 2 | The deforestation effect on temperature differences between neighboring days.** The probability density distribution of temperature differences between neighboring days ($\delta T$) for the piControl (blue bars) and deforest-globe (orange bars) simulations in the deforested areas of **a**–**d** North America and **e**–**h** Eurasia (denoted by the black rectangles in Fig. 1c) in **a**, **e** DJF (December, January and February), **b**, **f** MAM (March, April and May), **c**, **g** JJA (June, July and August), and **d**, **h** SON (September, October and November). The $p$ value denotes the statistical significance of the difference in the distribution of $\delta T$ between piControl and deforest-globe tested by the Kolmogorov–Smirnov two-sample test. The frequency changes for the near-zero ($-3\,°C < \delta T < 3\,°C$), left tail ($\delta T < -3\,°C$) and right tail ($\delta T > 3\,°C$) regions are shown at the top of each panel. The green line shows the percentage change $\left(\frac{\text{deforest-globe} - \text{piControl}}{\text{piControl}} \times 100\%\right)$ in the frequency for each $\delta T$ bin due to deforestation.

deforested areas in North America (38–54°N, 60–122°W) and Eurasia (52–66°N, 20–120°E; denoted by the black rectangles in Fig. 1c). Compared to the piControl simulation (blue bars), the frequency is lower for the near-zero region (e.g., $-3\,°C < \delta T < 3\,°C$) and higher for the tail regions (e.g., $\delta T < -3\,°C$ or $\delta T > 3\,°C$) for the deforest-globe simulation (orange bars). Meanwhile, the frequency changes for the left and right tail regions are almost identical. Taking North America as an example, the frequency for wintertime $\delta T$ within $-3\,°C$ to $3\,°C$

decreases by 5.3%, whereas the frequency for wintertime $\delta T$ below $-3\,°C$ and above $3\,°C$ increases by 2.8% and 2.5%, respectively (Fig. 2a). This result implies an increased likelihood of both rapid cooling and warming events following deforestation. Moreover, the difference in the probability density distribution of $\delta T$ between piControl and deforest-globe is statistically significant at the 95% confidence level (see the $p$ value in Fig. 2) tested by the Kolmogorov–Smirnov two-sample test (see Methods), except for Eurasia in summer.

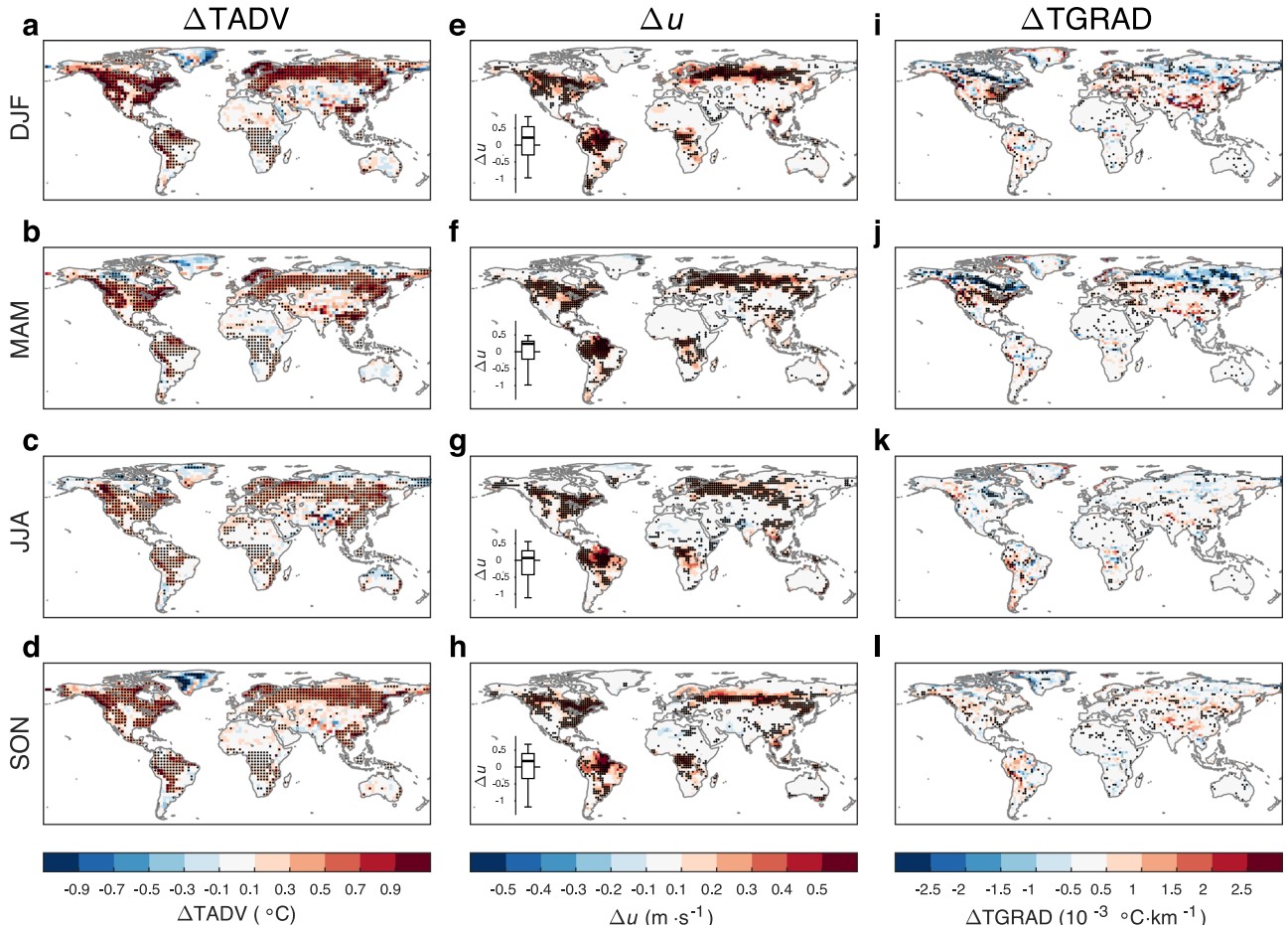

**Fig. 3 | The deforestation effect on near-surface horizontal temperature advection and its components.** The differences in near-surface horizontal temperature advection (ΔTADV), wind speed (Δ$u$), and the magnitude of the horizontal temperature gradient (ΔTGRAD) in **a**, **e**, **i** DJF (December, January and February), **b**, **f**, **j** MAM (March, April and May), **c**, **g**, **k** JJA (June, July and August), and **d**, **h**, **l** SON (September, October and November) between the piControl and deforest-globe simulations (deforest-globe minus piControl). ΔTADV is from the CanESM5 model, and the black dots in **a**–**d** indicate that ΔTADV is statistically significant at the 95% confidence level tested by Student's $t$-test. Δ$u$ and ΔTGRAD are from the multi-model mean, and the black dots in **e**–**l** indicate that all five models agree on the sign of Δ$u$ or ΔTGRAD. The box-and-whisker plots embedded in **e**–**h** show the wind speed differences between the paired forest and openland sites (openland minus forest; Supplementary Fig. 5) from the FLUXNET and AmeriFlux datasets. The horizontal black line of the box denotes the median. The bottom and top edges of the box denote the 25th and 75th percentiles, respectively. The whiskers extend to the 10th and 90th percentiles.

Intuitively, the deforestation-induced increase in the frequency for the tail regions seems small (no more than 6%). Given the low frequency for the tail regions, however, such an increase is not trivial. We further examine the percentage change ($\frac{\text{deforest}-\text{globe}-\text{piControl}}{\text{piControl}} \times 100\%$) in the probability density distribution of $\delta T$ following deforestation (see the green lines in Fig. 2). The percentage change in the frequency for the tail regions is considerable and nonmonotonically increases with higher magnitudes of $\delta T$. For example, the frequencies of moderate ($-10\,°C < \delta T < -5\,°C$ or $5\,°C < \delta T < 10\,°C$) and extreme ($\delta T < -10\,°C$ or $\delta T > 10\,°C$) wintertime $\delta T$ increase by 12.8% and 34.1%, respectively, in North America. Note that such percentage changes in the frequency for the tail regions can be even larger at local scales (Supplementary Fig. 6). This amplified increase in the frequency for the tail regions implies larger influences of deforestation on more extreme rapid warming or cooling events.

**The mechanisms for the deforestation effect on daily temperature variability**

Explaining the change in temperature variability is always challenging, and we attempt to reveal the mechanisms for the deforestation effect on DTDT based on the thermodynamic energy equation[54]. Following this equation, an increase in DTDT can be attributed to (1) an increase in near-surface horizontal TADV and/or (2) an increase in the day-to-day variation in surface sensible heat flux (DTDSHF; see Methods).

Figure 3a–d shows the TADV response to deforestation for each season. The CanESM5 result is shown here because only this model provides the required daily 10-m wind speed to calculate the daily TADV. The CanESM5 model suggests a widespread increase in TADV in most deforested areas and neighboring regions (Fig. 3a–d). The TADV signal is stronger (>0.5 °C) in the northern extratropical deforested areas, with the largest magnitude in winter, followed by spring and autumn, and the lowest magnitude in summer. However, the TADV signal is mostly weaker (within 0.3 °C) in the deforested areas in the tropics and the Southern Hemisphere, except for some marginal areas of the Amazon basin and the Indo–China Peninsula, throughout the year.

An increase in TADV can be further attributed to an increase in near-surface wind speed or horizontal temperature gradient (see Methods). Figure 3e–l shows the multimodel mean responses of near-surface wind speed and horizontal temperature gradient to deforestation for each season. The models (Fig. 3e–h) consistently suggest an increase in wind speed due to the lower surface roughness in deforested areas. This result is also supported by the paired site observations (see the box-and-whisker plots imbedded in Fig. 3e–h),

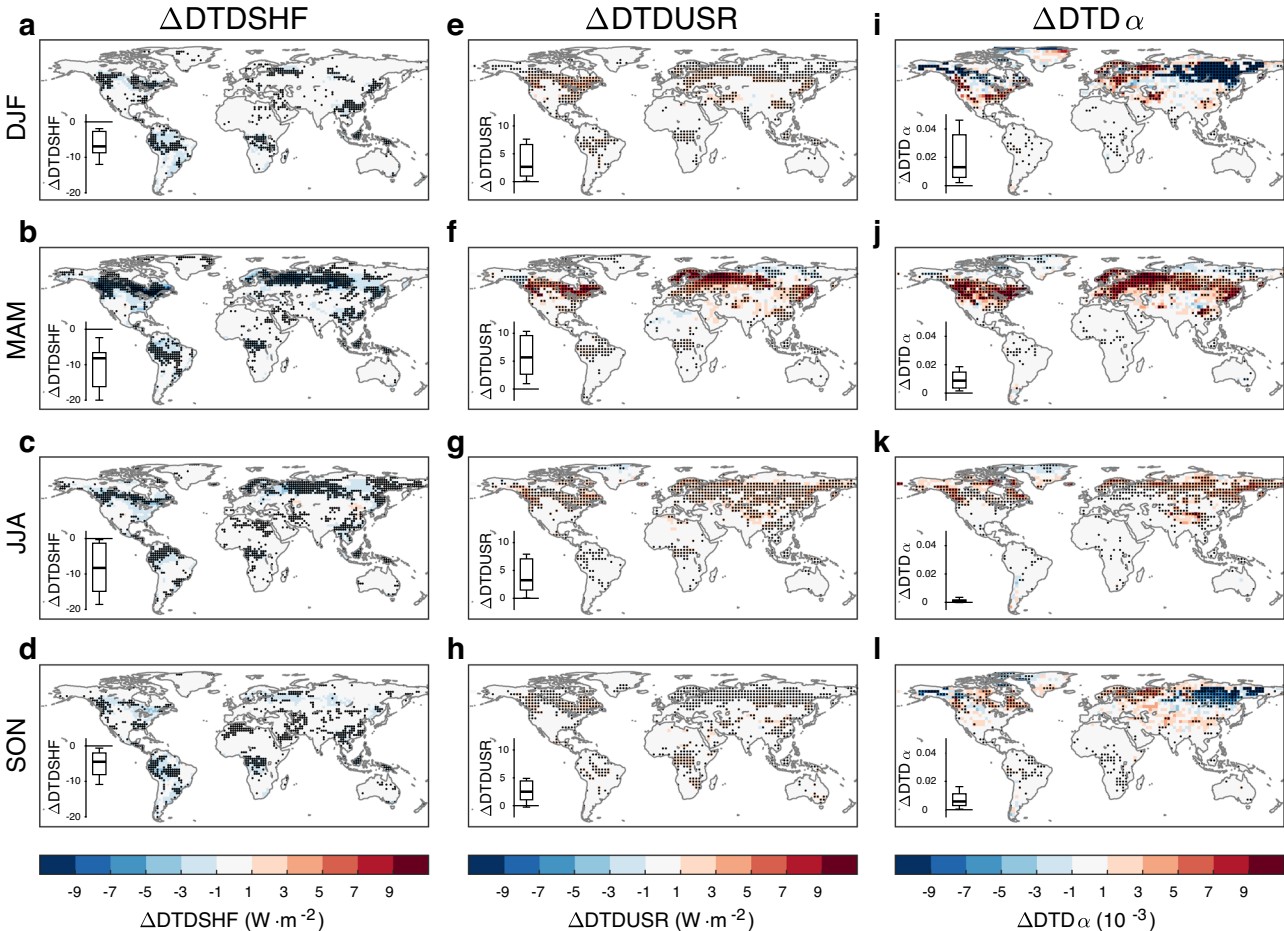

**Fig. 4 | The deforestation effect on daily variabilities in surface sensible heat flux, upward shortwave radiation, and albedo.** The differences in daily variabilities in surface sensible heat flux (ΔDTDSHF), upward shortwave radiation (ΔDTDUSR), and albedo (ΔDTDα) in **a**, **e**, **i** DJF (December, January and February), **b**, **f**, **j** MAM (March, April and May), **c**, **g**, **k** JJA (June, July and August), and **d**, **h**, **l** SON (September, October and November) between the piControl and deforest-globe simulations (deforest-globe minus piControl). ΔDTDSHF is from the multimodel mean, and the black dots in **a**–**d** indicate that all five models agree on the sign of

ΔDTDSHF. ΔDTDUSR and ΔDTDα are from the CanESM5 model, and the black dots in **e**–**l** indicate that ΔDTDUSR or ΔDTDα is statistically significant at the 95% confidence level tested by Student's *t*-test. The box-and-whisker plot embedded in each panel shows the corresponding difference between the paired forest and openland sites (openland minus forest; Supplementary Fig. 5) from the FLUXNET and AmeriFlux datasets. The horizontal black line of the box denotes the median. The bottom and top edges of the box denote the 25th and 75th percentiles, respectively. The whiskers extend to the 10th and 90th percentiles.

which show overall higher wind speeds in openland sites than in neighboring forest sites. The models also consistently suggest an increase in temperature gradient in some midlatitude deforested areas in winter and spring (Fig. 3i–l). This enhanced temperature gradient can be explained by the strong cooling effect of deforestation in boreal regions, which increases the north-south temperature difference. The increased wind speed and temperature gradient collaborate to enlarge the TADV in the northern extratropical deforested areas.

Figure 4a–d shows the multimodel mean DTDSHF response to deforestation for each season. The models consistently suggest a year-round decrease in DTDSHF in almost all deforested areas (Fig. 4a–d). The DTDSHF change is larger (up to −9 W·m⁻²) in spring and summer but small (mostly within −3 W·m⁻²) in winter and autumn in the northern extratropical deforested areas. For the tropical deforested areas, however, the DTDSHF change is moderate (−5 to −1 W·m⁻²) and stable throughout the year. In line with the simulations, the paired site observations also show lower DTDSHFs in openland sites than in neighboring forest sites. Such a decrease in DTDSHF can be explained by the decreased surface aerodynamic roughness following deforestation. The lower roughness causes the surface to be less efficient in warming the aboveground atmosphere through turbulence, suppressing the daily variability in SHF. The daily variability of SHF can be

alternatively quantified by the standard deviation index, and a similar result can be obtained (not shown).

To explain the seasonal variation in the DTDSHF response to deforestation, we further examine the daily variabilities in the other components (including shortwave and longwave radiations and latent heat flux) of the surface energy balance based on the CanESM model given data availability. As shown in Fig. 4e–l, the daily variabilities in surface upward shortwave radiation (DTDUSR) and albedo (DTDα) increase in deforested areas. This result indicates that deforestation enlarges not only the surface albedo and the reflected solar radiation, but also their variabilities. In the northern extratropic deforested areas, the DTDUSR change is larger in spring because of the larger DTDα change (due to the snow cover) as well as the higher springtime solar radiation. The larger DTDUSR change mostly explains the larger DTDSHF change in the northern extratropic deforested areas in spring. Moreover, the simulated responses of DTDSUR and DTDα to defor-estation are also supported by the paired site observations (see the box-and-whisker plots embedded in Fig. 4e–l). The daily variabilities in surface net longwave radiation and latent heat flux are less influenced by deforestation (not shown).

The DTDSHF and TADV changes can basically explain the DTDT response to deforestation. For the northern extratropical deforested

areas, the enhanced TADV, despite being slightly offset by the decreased DTDSHF, dominates the DTDT increase. Meanwhile, the TADV signal is stronger in winter, spring and autumn, whereas the DTDSHF signal is stronger in spring and summer, causing seasonal variations in the DTDT response to deforestation there. For example, the larger increase in wintertime DTDT is a consequence of the larger increase in wintertime TADV and the small change in wintertime DTDSHF. For the tropical deforested areas, however, the TADV change is small and mostly offset by the decreased DTDSHF. Moreover, the climatological mean DTDT is smaller in the tropics than in boreal regions. As a result, the DTDT change is negligibly small in the tropical deforested areas.

As mentioned earlier, the observed DTDT signal from the paired sites is inconsistent with the simulated signal in spring and summer (Fig. 1d, e). This discrepancy can be explained by the TADV response. Specifically, the atmosphere above the paired forest and openland sites tends to be mixed due to horizontal advection, alleviating the TADV difference between the paired sites throughout the whole year. Thus, the DTDT differences between the paired sites are more influenced by the DTDSHF effect, particularly in spring and summer when the TADV effect is smaller (inferred from the simulations) but the DTDSHF effect is larger (Fig. 4a–d). That is why the observations suggest an overall lower DTDT in openland sites in spring and summer, contrasting with the simulated DTDT change.

### The effect of historical deforestation on daily temperature variability

The above analysis is built on the idealized deforestation scenario, which is unlikely to occur in the real world. We therefore wonder whether the real-world deforestation over the historical period would have generated a detectable signal as well. To this end, the historical simulation (historical) of CMIP6 and the historical no land-use simulation (hist-noLu) of LUMIP are used. historical covers the period of 1850–2014 with evolving and externally imposed natural (e.g., solar and volcanic aerosols) and anthropogenic (e.g., GHGs, aerosols, and land use/land cover changes (LULCCs)) forcings (see Methods). hist-noLu is identical to historical, except that LULCCs are fixed at preindustrial levels (see Methods). The last 30-year simulations (1985–2014) of historical and hist-noLu are compared (historical minus hist-noLu) to isolate the deforestation effect. Nine earth system models are used here given the model output availability (Supplementary Table S1). These models are also reasonable in the DTDT simulation validated against the reanalysis (Supplementary Fig. 1e–h, i–p; see Methods).

Tree cover changes during the historical period are highly heterogeneous in space with the concurrence of forest losses and gains (Fig. 5a), which may confound the signals arising from forest losses and gains. Exceptionally, we find tremendous and spatially coherent tree cover losses in North America since 1850 (Fig. 5a). Moreover, this decreasing trend in tree cover is consistent across the models (Supplementary Fig. 7), with the multimodel mean value of the net tree cover loss being approximately −1.3 million km² during the historical period of 1850–2014. As a result of such large-scale deforestation, seven out of the eight models simulate increases in wintertime DTDT (with multimodel mean values of 0.2–0.5 °C or 2–12%; Fig. 5b) and SDT (with multimodel mean values of 0.2–0.6 °C or 2–10%; not shown). In other words, historical deforestation has led to a detectable increase in the wintertime daily temperature variability in North America. The enhanced daily temperature variability can be attributed to the higher TADV (Fig. 5d) and simultaneously is slightly offset by the lower DTDSHF (Fig. 5c). Although the TADV change is from a single model (ACCESS-ESM1-5) due to data availability, these mechanisms are consistent with those for the effect of idealized deforestation. The consistent DTDT responses to the idealized deforestation scenario and the historical deforestation scenario as well as the consistent mechanisms consolidate the robustness of the results.

Note that there is some spatial mismatch between the DTDT change (Fig. 5b) and the TADV change (Fig. 5d) in North America. This mismatch is because the pattern of tree cover changes for the ACCESS-ESM1-5 model (Supplementary Fig. 7a) is slightly different from the multimodel mean pattern (Fig. 5a). Examining the ACCESS-ESM1-5 model, we find that the DTDT and TADV changes from this model overall match each other in terms of the geographical pattern (Fig. 5d and Supplementary Fig. 8).

We further examine whether the deforestation effect on daily temperature variability is noteworthy compared to the effects of other anthropogenic forcings (e.g., GHGs and aerosols). To this end, the historical natural-only simulation (hist-nat) of the Detection and Attribution Model Intercomparison Project[55] is used. hist-nat is identical to historical but is imposed by only natural forcings (see Methods). Accordingly, comparing the last 30-year (1985–2014) simulations of historical and hist-nat (historical minus hist-nat) isolates the net effect of all anthropogenic forcings (GHGs, aerosols, and LULCCs) during the historical period. Averaged over global land, the deforestation effect on DTDT (0.01 °C) is much smaller than the combined effect of GHGs and aerosols (−0.03 °C) in magnitude (Fig. 5f). At local and regional scales, however, the deforestation effect can be comparable to the combined effect of GHGs and aerosols in magnitude, particularly in deforested areas. In North America, for example, the wintertime DTDT shows an overall decreasing trend during the historical period with all anthropogenic forcings considered (Fig. 5e). This decreasing trend in daily temperature variability is also supported by previous studies[34–37]. Interestingly, this decreasing trend in DTDT is mostly absent in the deforested area of North America (Fig. 5e), implying that the positive effect of deforestation (Fig. 5c) largely offset the combined effect of GHGs and aerosols. Averaged over the deforested area of North America, the contributions of deforestation and the combined GHG and aerosol forcings to the historical DTDT change are 0.11 and −0.17 °C, respectively (Fig. 5f).

### The effect of potential afforestation on daily temperature variability

Based on the above results, we speculate that future afforestation in the northern extratropics may reduce daily temperature variability. To test this speculation, the scenario simulation (ssp370) of the Scenario Model Intercomparison Project[55] and the future land-use policy sensitivity simulation (ssp370-ssp126Lu) of the LUMIP are used. ssp370 covers the future period of 2015–2100 under the shared socioeconomic pathway 3-7.0 scenario (SSP3-7.0) with substantial deforestation and high emissions (see Methods). ssp370-ssp126Lu is identical to ssp370 except that LULCCs are from the shared socioeconomic pathway 1-2.6 scenario (SSP1-2.6) with substantial afforestation or reforestation. The last 30-year simulations (2071–2100) of ssp370 and ssp370-ssp126Lu are compared (ssp370-ssp126Lu minus ssp370) to isolate the afforestation effect. Six earth system models are used here given the model output availability (Supplementary Table S1).

Figure 6a shows the differences in tree cover between the SSP1-2.6 and SSP3-7.0 scenarios by the end of the twenty-first century. Benefiting from afforestation or reforestation anticipated by the SSP1-2.6 scenario, we can see increases in tree cover in central Africa, eastern America, the western Amazon, western Europe, and central China. Figure 6b–e shows the biogeophysical effect of these tree cover changes on DTDT for each season. As expected, the models consistently project that the regional DTDT will be reduced by 0.06–0.1 °C in winter, spring and autumn in eastern America by the end of the twenty-first century. For western Europe and central China, however, the afforestation effect on DTDT is mostly small, probably due to the small increase in tree cover anticipated by the SSP1-2.6 scenario (mostly within 10%). While the tree cover changes in central Africa and the western Amazon are considerable, the effect on DTDT is also limited throughout the year. This small DTDT response can be attributed

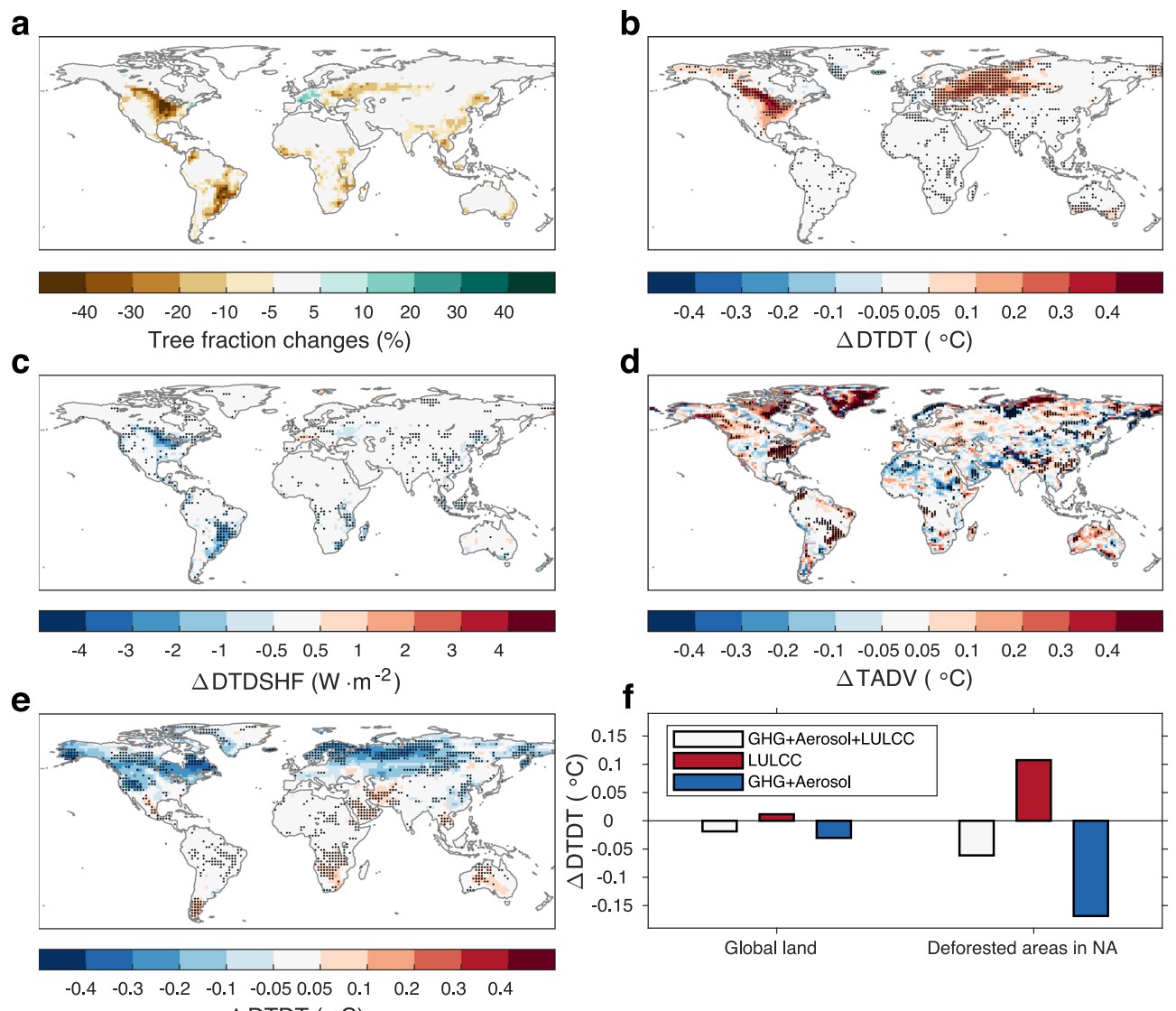

**Fig. 5 | Historical tree cover changes and their effects on wintertime daily temperature variability. a** Multimodel mean total tree fraction differences between the historical and hist-noLu simulations (historical minus hist-noLu) during the historical period (1850–2014). The biogeophysical effect of historical tree cover changes on **b** day-to-day temperature variability (DTDT), **c** day-to-day variability in sensible heat flux (DTDSHF), and **d** the near-surface horizontal temperature advection (TADV) in DJF (December, January and February). The DTDT and DTDTSHF changes are from the mean value of eight models, and the black dots in **b** and **c** indicate that at least seven out of the eight models agree on the sign of the change. The TADV change is from the ACCESS-ESM1-5 model, and the black dots in **d** indicate that the TADV change is statistically significant at the 95% confidence level tested by Student's *t*-test. **e** The net effects of all anthropogenic forcings, including greenhouse gases (GHGs), aerosols, and land use/land cover changes (LULCCs), on DTDT in DJF during the historical period. In panel **e**, the DTDT change is from the mean value of four models, and the black dots indicate that all four models agree on the sign of the change. **f** The individual contributions of anthropogenic forcings to the historical DTDT change averaged over global land and the deforested area (total tree cover loss >5%) of North America.

to the low sensitivity of DTDT to forest changes in these tropical regions (Fig. 1c–f).

## Discussion

Much attention has been given to the biogeophysical effects on the mean, the diurnal cycle, or extremes of temperature, whereas the effect on temperature variability has rarely been studied. Here, we show that the biogeophysical effect of deforestation on daily temperature variability is considerable. The higher daily temperature variability can further lead to more frequent rapid warming and cooling events. In particular, the deforestation effect on extreme warming and cooling effects (e.g., the temperature difference between neighboring days exceeding 10 °C) are larger, implying more frequent extreme weather events, e.g., hot and cold waves. We also find that the

deforestation effect on daily temperature variability can be comparable to the effects of other anthropogenic forcings in magnitude at local and regional scales. Therefore, this study highlights that the biogeophysical effect of deforestation should also be considered when human contributions to the trend of daily temperature variability are quantified.

In contrast to the deforestation effect, afforestation can reduce the daily temperature variability in the northern extratropics. Since changes in daily temperature variability have broad impacts on humans[21–25,30] and ecosystems[26–29], our results have implications for the implementation of large-scale afforestation or reforestation programs, particularly for northern extratropical countries. In these regions, whether afforestation should be implemented remains controversial. On the one hand, afforestation is recommended because it can provide

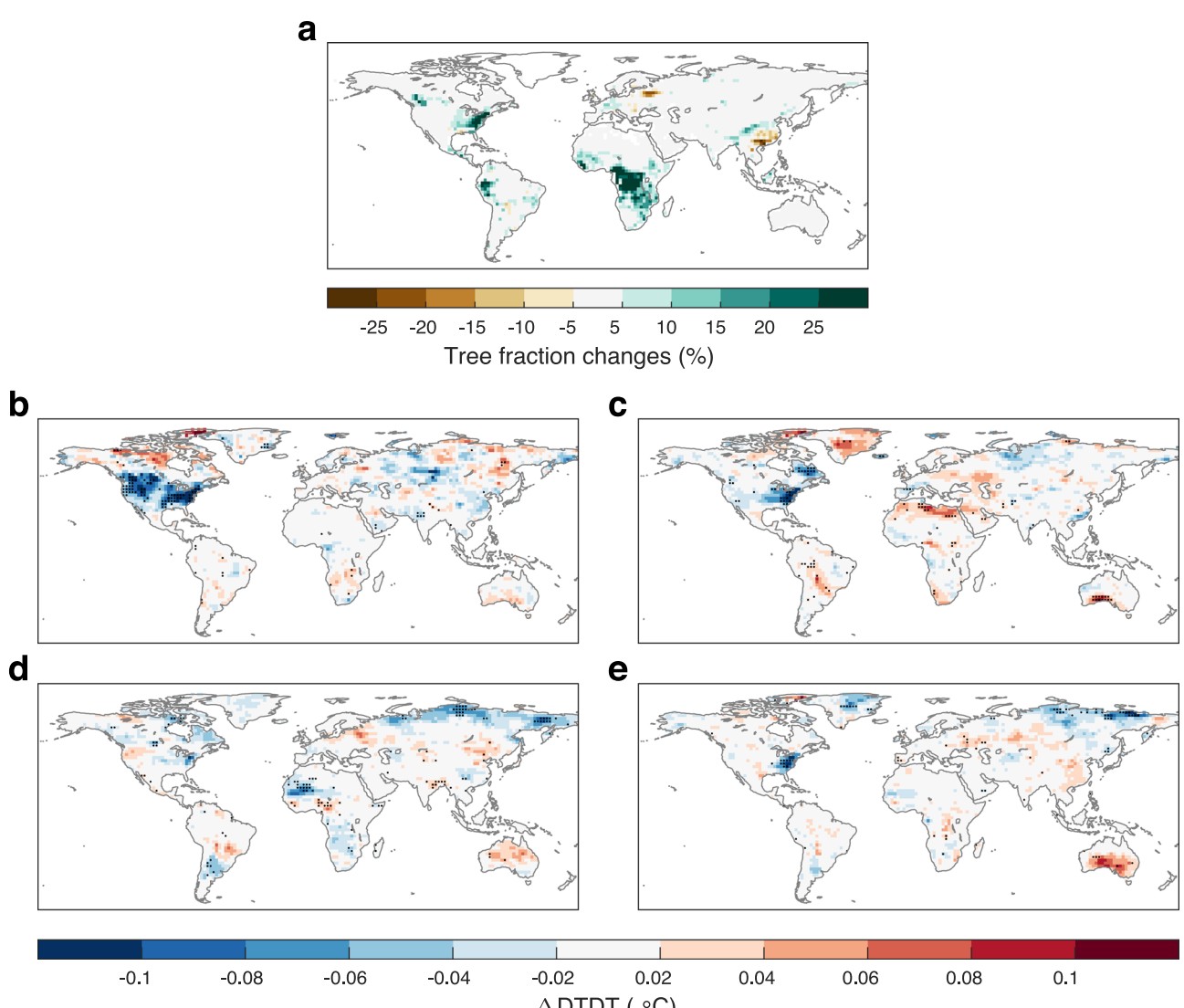

**Fig. 6 | Potential tree cover changes by the end of the twenty-first century and their effects on daily temperature variability. a** Multimodel mean total tree fraction differences between the ssp370 and ssp370-ssp126Lu simulations (ssp370-ssp126Lu minus ssp370) by the end of the twenty-first century (2071–2100). **b**–**e** The biogeophysical effect of potential tree cover changes on day-to-day temperature variability (DTDT) in **b** DJF (December, January and February), **c** MAM (March, April and May), **d** JJA (June, July and August), and **e** SON (September, October and November). The black dots in **b**–**e** denote that all six models agree on the sign of the DTDT change.

invaluable ecological, economic, social, and aesthetic services[3]. Afforestation is also expected to mitigate climate warming through the carbon sequestration effect[56] and to alleviate the risks of hot extremes through higher heat dissipation and evaporative efficiency[16,19,20]. On the other hand, afforestation is not recommended, as it may fail to mitigate climate warming due to albedo-driven warming effects[43,57,58]. In addition to the afforestation effect on the carbon cycle and temperature mean and extremes discussed in previous studies, we call for further taking the afforestation effect on daily temperature variability into consideration when large-scale afforestation or reforestation is implemented in northern extratropical countries.

We identify the enhanced near-surface horizontal wind speed and TADV as the main drivers of the higher daily temperature variability over deforested areas. It should be emphasized that the response of wind to deforestation may depend on the deforestation scale. In this study, the enhanced wind speed and TADV should be regarded as the consequence of large-scale deforestation. However, small-scale deforestation favors the generation of mesoscale circulations due to the higher SHF[59,60]. Such thermodynamically driven mesoscale circulations may further regulate the wind and TADV over deforested areas.

Moreover, the model resolution has also been proven to play an important role in determining the simulated response of mesoscale circulations to surface perturbations[61,62]. In particular, the high-resolution (<4 km) simulation with an explicit treatment of convection and the coarse-resolution (~10 to ~100 km) simulation with parameterized convection can even produce different responses of mesoscale circulations to the same surface perturbation[61]. In this study, only coarse-resolution earth system models are used, and the results are probably model resolution dependent. Therefore, high-resolution simulations (e.g., the convection-permitting simulation) are encouraged to further investigate the deforestation effect on daily temperature variability, particularly the effect of small-scale deforestation.

We acknowledge the uncertainty in the mechanisms for the biogeophysical effects of deforestation on daily temperature due to the model output availability. In particular, the effect related to horizontal near-surface TADV is examined based on limited models. It should be emphasized that, despite this limitation, the deforestation effect on daily temperature variability is robust because the simulations and observations consistently agree on the effect, and the results are independent of the metrics of the daily temperature variability. Since

the daily variables are required to study the daily temperature variability, we encourage more model groups that contribute to the CMIP6 and LUMIP to report daily output to support further analysis.

## Methods

### Investigation of the biogeophysical effect of the idealized deforestation

To obtain the biogeophysical effect of the idealized deforestation, the piControl simulation of the CMIP6[44] and the idealized global deforestation simulation (deforest-globe) of the LUMIP[44] are used.

piControl is a long-term (≥500 years) fully coupled simulation and serves as the baseline experiment of CMIP6. piControl is representative of the period prior to the onset of large-scale industrialization, with the reference year being 1850. Volcanic aerosols are fixed at, as closely as possible, the mean values over the historical period of 1850–2014. Solar variability is fixed at the mean value over the first two solar cycles of the historical period (e.g., 1850–1873). Anthropogenic forcings, e.g., GHGs, aerosols, and LULCCs, are fixed at the levels of the year 1850.

deforest-globe is branched from piControl and is an 80-year fully coupled simulation. In deforest-globe, 20 million $km^2$ of tree cover is linearly converted to natural grassland cover during the first 50 years. Deforestation is only performed on the top 30% grid cells with the highest initial fractional tree cover (determined by piControl). In the following 30 years, the tree cover is maintained at a constant value to allow the system to equilibrate. All external forcings except the land cover are the same as in piControl. While the response of terrestrial $CO_2$ flux to deforestation is allowed, the atmospheric $CO_2$ concentration is fixed at the value of piControl. In other words, deforestation-induced $CO_2$ emissions do not reach the atmosphere. Thus, deforestation only influences climate through biogeophysical processes.

The last 30-year (year 51–80) simulations since the branching year are used for analysis for piControl and deforest-globe. The biogeophysical effect of the idealized deforestation is obtained by comparing the 30-year simulations of deforest-globe and piControl (deforest-globe minus piControl). By the time of this study, five earth system models contribute to both the piControl and deforest-globe simulations and provide the required daily 2-meter temperature outputs; thus, these five models are used (Supplementary Table S1). For each model, only one member is available for deforest-globe; thus, the single member of deforest-globe and the corresponding member of piControl are used for analysis.

The total tree cover change is obtained by subtracting the tree cover fraction of piControl from the value at year 51 of deforest-globe. Following the same deforestation protocol of deforest-globe, the models are required to produce a spatial deforestation signal that, as closely as possible, replicates that shown in Fig. 1a. Nevertheless, the models still slightly differ in the tree cover changes at regional scales. This intermodel consistency is inevitable and can be attributed to the different initial tree cover fractions and divergent model structures, e.g., vegetation dynamics[45]. For the models incorporating the dynamic vegetation module, vegetation dynamics are required to be switched off (if possible) on the deforested grid cells and allowed outside the deforested grid cells[45]. The tree cover of EC-Earth3-Veg and UKESM1-0-LL still evolve during the last 30-year simulation (Fig. 1b) because the vegetation dynamics are switched on outside the deforested grid cells for these two models. Nevertheless, the tree cover change is limited during the last 30-year simulation for these two models.

### Investigation of the biogeophysical effect of historical deforestation

To obtain the biogeophysical effect of deforestation during the historical period, the historical simulation (historical) of CMIP6[44] and the historical no land-use simulation (hist-noLu) of LUMIP[45] are used.

historical is a fully coupled simulation covering the historical period of 1850–2014. This simulation is branched from piControl and

is driven by evolving and externally imposed forcings. All natural (e.g., volcanic aerosols and solar variability) and anthropogenic (e.g., GHGs, aerosols, and LULCCs) forcings are included and are largely based on observations. The historical LULCCs are prescribed by the second generation of the Land-Use Harmonization dataset[63] and include detailed changes in land cover, land use, and land management.

hist-noLu is the same as historical except with LULCCs fixed at the levels of the year 1850. Note that the atmospheric $CO_2$ concentration in hist-noLu is identical to the value in historical. Thus, the biogeophysical effect of historical deforestation is isolated by comparing the last 30-year (1985–2014) simulations of historical and hist-noLu (historical minus hist-noLu). By the time of this study, nine earth system models contribute to both the historical and hist-noLu simulations and provide the required daily 2-meter temperature output; thus, these nine models are used here (Supplementary Table S1). For each model, multiple members are available for historical and hist-noLu. To save computational cost, the first members of historical and hist-noLu are used for analysis.

The total tree cover change during the historical period is obtained by subtracting the last 30-year (1985–2014) mean tree cover fraction of hist-noLu from the corresponding value of historical. Note that while the historical LULCCs are prescribed by the Land-Use Harmonization dataset, the implementation of the historical LULCCs across different models is commonly different (Supplementary Fig. 7), and this problem has always plagued model intercomparison[45]. Such intermodel inconsistency may cause the models to diverge in the response of daily temperature variability to the historical LULCCs. Moreover, both tree losses and gains may occur in a region, confounding the signals in daily temperature variability arising from forest losses and gains. However, we identify a large area of forest losses during the historical period in North America (Fig. 5a). This tree cover change is considerable, spatially coherent, and of high intermodel consistency (Supplementary Fig. 7). Therefore, we take North America as a case study to investigate the response of daily temperature variability to historical deforestation.

### Investigation of the net effect of anthropogenic forcings

To obtain the net effect of anthropogenic forcings during the historical period of 1850–2014, the hist-nat simulation of the Detection and Attribution Model Intercomparison Project[64], as well as the historical, are used.

hist-nat resembles historical but is forced with only solar and volcanic forcings from historical. Thus, the net effect of all anthropogenic forcings (GHGs, aerosols, and LULCCs) is isolated by comparing the last 30-year (1985–2014) simulations of historical and hist-nat (historical minus hist-nat). By the time of this study, four earth system models contribute to both the historical and hist-nat simulations and provide the required daily 2-meter temperature output; thus, these four models are used here (Supplementary Table S1). For each model, multiple members are available for historical and hist-nat. Similarly, the first members of historical and hist-nat are used for analysis.

### Investigation of the biogeophysical effect of potential afforestation

To obtain the biogeophysical effect of potential afforestation by the end of the twenty-first century, the ssp370 simulation of the Scenario Model Intercomparison Project[55] and the ssp370-ssp126Lu simulation of the LUMIP are used.

ssp370 is a fully coupled simulation covering the future period of 2015–2100 under the SSP3-7.0 scenario. This scenario represents the medium to high end of the range of future forcing pathways with substantial LULCCs (in particular global forest losses) and high emissions of near-term climate forcers (namely, tropospheric aerosols, tropospheric $O_3$ precursors, and $CH_4$). ssp370-ssp126Lu is the same as

ssp370, but LULCCs are from the SSP1-2.6 scenario. The SSP1-2.6 scenario represents the low end of the range of future forcing pathways that aim to constrain the global mean temperature below 2 °C by 2100. In contrast to the SSP3-7.0 scenario, there are substantial forest gains due to afforestation or reforestation for the SSP1-2.6 scenario. The LULCCs for both SSP1-2.6 and SSP3-7.0 are prescribed by the Land-Use Harmonization dataset. Note that the atmospheric $CO_2$ concentration in ssp370 is identical to the value in ssp370-ssp126Lu. Thus, the biogeophysical effect of potential afforestation is isolated by comparing the last 30-year (2071–2100) simulations of ssp370 and ssp370-ssp126Lu (ssp370-ssp126Lu minus ssp370).

By the time of this study, six earth system models contribute to both the ssp370 and ssp370-ssp126Lu simulations and provide the required daily 2-meter temperature output; thus, these six models are used here (Supplementary Table S1). For each model, only one member is available for ssp370-ssp126Lu. Thus, the single member of ssp370-ssp126Lu and the corresponding member of ssp370 are used for analysis.

## Definitions of the daily temperature variability

To quantify the daily temperature variability, two indices are used: the DTDT[45] and the SDT[32]. Specifically, DTDT is defined as the mean value of absolute differences in temperature between every two neighboring days during a given time period, expressed as:

$$DTDT = \frac{1}{n-1}\sum_{i=1}^{n-1}|T_{i+1} - T_i| \tag{1}$$

where $T_i$ denotes the 2-meter temperature on day $i$, and $n$ denotes the total days of the period.

SDT is a more commonly used index to quantify daily temperature variability and is expressed as:

$$SDT = \sqrt{\frac{1}{n-1}\sum_{i=1}^{n}\left(T_i^* - \bar{T}\right)^2} \tag{2}$$

where $T_i^*$ denotes the 2-meter temperature on day $i$ with the annual cycle removed.

Note that DTDT and SDT differ in some aspects, although they both describe daily temperature variability[31,37,65]. First, SDT is insensitive to the positions of daily temperature anomalies. For example, an orderly (e.g., 25, 25, 25, 25, 15, 15, 15 and 15 °C) and an oscillatory (e.g., 25, 15, 25, 15, 25, 15, 25 and 15 °C) time series of daily temperature have identical SDT values (5.27 °C). However, the temperature changes between neighboring days can be felt once and seven times in the orderly and oscillatory series, respectively. Such a difference between the orderly and oscillatory series can be reflected by DTDT (1.43 versus 10 °C). Second, compared to DTDT, SDT contains variabilities at multiple time scales and may be susceptible to low-frequency temperature variabilities (e.g., weekly variability).

Here, we do not determine which index is better for describing daily temperature variability. Thus, both DTDT and SDT are used to quantify daily temperature variability. More importantly, the results are independent of the metrics of daily temperature variability (Fig. 1c–f versus Supplementary Fig. 4).

Some models provide the daily 2-meter mean temperature that can be used to directly calculate DTDT and SDT. A few models, however, only provide the daily 2-meter maximum and minimum temperatures. For these models, the maximum and minimum temperatures are averaged to produce the estimate of the mean temperature, and then the estimate is further used to calculate DTDT and SDT. The model output variables required to calculate DTDT and SDT are listed in Supplementary Table S2.

## Attribution of the change in daily temperature variability

According to the thermodynamic energy equation[54], the temperature difference ($\delta T$) between two neighboring days can be expressed as:

$$\delta T = \frac{\partial T}{\partial t} = -\mathbf{V} \cdot \nabla T + \left(\frac{RT}{c_p P} - \frac{\partial T}{\partial P}\right)\omega + \frac{1}{c_p}\frac{\partial Q}{\partial t} \tag{3}$$

where $T$ is the daily 2-meter temperature (°C), $\mathbf{V}$ is the daily 10-meter vector wind (m·s$^{-1}$), $\nabla$ is the horizontal divergence operator, $R$ is the gas constant for dry air (287 J·K$^{-1}$·kg$^{-1}$), $c_p$ is the specific heat of dry air at constant pressure (1004 J·K$^{-1}$·kg$^{-1}$), $P$ is the atmospheric pressure (Pa), $\omega$ is the daily near-surface vertical velocity (Pa·s$^{-1}$), and $Q$ is the daily diabatic heating (J·kg$^{-1}$).

Substituting Eq. (3) into Eq. (1), Eq. (1) can be rewritten as:

$$DTDT = \frac{1}{n-1}\sum_{i=1}^{n-1}|\delta T_i| = \frac{1}{n-1}\sum_{i=1}^{n-1}|\frac{\partial T}{\partial t}_i| \approx term1 + term2 + term3 \tag{4}$$

$$term1 = \frac{1}{n-1}\sum_{i=1}^{n-1}|-\mathbf{V}_i\cdot\nabla T_i| \tag{5}$$

$$term2 = \frac{1}{n-1}\sum_{i=1}^{n-1}|\left(\frac{RT_i}{c_p P_i} - \frac{\partial T}{\partial P_i}\right)\omega_i| \tag{6}$$

$$term3 = \frac{1}{n-1}\sum_{i=1}^{n-1}|\frac{1}{c_p}\frac{\partial Q}{\partial t}_i| \tag{7}$$

Then, the deforestation-induced change in DTDT can be expressed as:

$$\Delta DTDT = \Delta term1 + \Delta term2 + \Delta term3 \tag{8}$$

where $\Delta$ denotes the difference between the piControl and deforest-globe simulations (deforest-globe minus piControl) or between the historical and hist-noLu simulations (historical minus hist-noLu). $\Delta term1$ denotes the change in the horizontal near-surface TADV. $\Delta term2$ denotes the change in adiabatic compression and vertical advection. $\Delta term2$ is commonly negligibly small in magnitude and can be reasonably omitted[65]. $\Delta term3$ denotes the change in the daily variability of diabatic heating.

For $\Delta term3$, diabatic heating is considered to be determined by surface SHF because deforestation directly impacts near-surface temperature mainly through the change in SHF[5,6]. Strictly speaking, to calculate $\Delta term3$, the sensible heat absorbed by the 2-meter air should be used. However, only the total sensible heat that warms the aboveground air is known. To compromise, we use the total sensible heat to qualitatively estimate the contribution of $\Delta term3$ to the DTDT change, expressed as:

$$\Delta term3 = \Delta\left(\frac{1}{n-1}\sum_{i=1}^{n-1}|\frac{\partial SHF}{\partial t}_i|\right) = \Delta\left(\frac{1}{n-1}\sum_{i=1}^{n-1}|SHF_{i+1} - SHF_i|\right) \tag{9}$$

where $SHF_i$ denotes the surface SHF on day $i$. Obviously, $\Delta term3$ is actually the change in the DTDSHF (see Eq. (1)). It should be emphasized that this approach prevents us from showing $\Delta term3$ in a unit equivalent to $\Delta DTDT$. In other words, $\Delta term3$ (Eq. (9)) can only qualitatively explain the DTDT change, e.g., positive $\Delta term3$ leading to increased DTDT.

In summary, the DTDT response to deforestation can be attributed to the change in the horizontal near-surface TADV and the change in the DTDSHF. Note that the effects of changes in albedo, surface roughness, and evaporation efficiency that may arise from

deforestation have already been covered by the TADV and DTDSHF effects. For example, the albedo-driven cooling effect and the evapotranspiration-driven warming effect determine the surface temperature change, which directly influences DTDSHF and indirectly influences the TADV through the modification of the horizontal temperature gradient. The lower surface roughness increases near-surface wind, favoring the higher TADV. Similarly, the atmospheric feedbacks[66,67] that arise from deforestation to surface temperature (e.g., cloud formation and downwards radiation) are also implicitly included in the TADV and DTDSHF effects.

The model output variables required to calculate TADV and DTDSHF are listed in Supplementary Table S2.

## Validation of the model simulation of daily temperature variability

To validate the model behavior in the simulations of DTDT and SDT, two independent reanalysis datasets are used: the ERA5 reanalysis[46] from the European Center for Medium Weather Forecasting and the NCEP-DOE AMIP-II reanalysis[47] from the National Centers for Environmental Prediction and National Center for Atmospheric Research. These two reanalysis datasets can provide reliable observations of daily temperature variability[36] and are widely used in previous studies[36,38]. The ERA5 reanalysis provides the global daily mean 2-meter temperature at a horizontal resolution of 0.25° (latitude) × 0.25° (longitude) since 1950. The NCEP-DOE AMI-II reanalysis provides global daily mean 2-meter temperature on the T62 Gaussian grid (192 longitude grids × 94 latitude grids) since 1979.

The 30-year (1985–2014) mean DTDT and SDT from the ERA5 reanalysis (Supplementary Figs. 1i–l and 3i–l) and the NCEP-DOE AMIP-II reanalysis (Supplementary Figs. 1m–p and 3m–p) are used as the observational benchmarks. The 30-year (year 51–80) mean DTDT and SDT simulated by piControl (Supplementary Figs. 1a–d and 3a–d) and the 30-year mean (1985–2014) DTDT and SDT simulated by historical (Supplementary Figs. 1e–h and 3e–h) are validated against the reanalysis. Note that the anthropogenic forcings for piControl are fixed at the levels of the year 1850, which are inconsistent with the reanalysis levels. Thus, there might be larger biases in DTDT and SDT simulated by piControl than historical.

Validated with the reanalysis, both piControl and historical can reasonably simulate the global patterns of DTDT and SDT for each season. Statistically speaking, compared to the ERA5 reanalysis, the root mean squared error in the simulated DTDT is between 0.17 and 0.35 °C for piControl and between 0.16 and 0.24 °C for historical across the seasons. When compared to the NCEP-DOE AMIP-II reanalysis, the root mean squared error in DTDT is slightly larger for both piControl (0.25–0.37 °C across the seasons) and historical (0.23–0.47 °C across the seasons). In general, the root mean squared error in the simulated DTDT is highest in winter, followed by spring and autumn, and lowest in summer. A similar result can be obtained for the simulated SDT.

## Validation of the model representation of the biogeophysical effect of deforestation on mean surface temperature

To evaluate the model behavior in the representation of the biogeophysical effect of deforestation on mean surface temperature, a dataset[48] mapping the biophysical effect of vegetation cover change at the global scale is used. This dataset is established based on the "space-for-time" substitution method and multiple satellite observations. We use the dataset mapping the potential changes in daytime (-13:30 local time) and nighttime (-01:30 local time) surface temperatures as a result of a complete transition from forests to grasslands and croplands. The dataset is provided at monthly time steps and a horizontal resolution of 1° × 1°. The monthly values are averaged over the period of 2008–2012. The daytime and nighttime values are averaged to obtain the estimate of the daily mean value.

Compared to the satellite observations, the models simulate more widespread cooling effects in the northern mid-latitudes (Supplementary Fig. 2). It should be emphasized that such discrepancies cannot be fully attributed to the poor performance of the models. Instead, this discrepancy largely results from the unfair comparison between the observed and simulated biogeophysical effects[50]. Specifically, the observed biogeophysical effect of deforestation is retrieved based on the "space-for-time" substitution method through the comparison of the spatially adjacent forest and openland pixels. Since the atmosphere is assumed to be largely identical over forest pixels and neighboring openland pixels, the deforestation-induced atmospheric feedbacks to surface temperature (e.g., cloud formation and downwards radiation)[66,67] are not fully considered. Thus, the observed biogeophysical effect should be considered largely representative of the local effect of deforestation[15,48]. In contrast, the atmospheric feedbacks to deforestation are fully considered in the coupled models. Thus, the simulated biogeophysical effect includes both the local and nonlocal (through atmospheric feedback) effects of deforestation[49]. It has been explicitly demonstrated that deforestation causes widespread non-local cooling effects in the northern mid- to high-latitudes, with magnitudes comparable to or even larger than the local effects[49,68]. That is why the simulations indicate a more widespread cooling effect of deforestation in the northern mid-latitudes than the satellite observations[49,50] (Supplementary Fig. 2). Moreover, the biogeophysical effect of deforestation also depends on the background climate[69,70]. Thus, the cold bias in the simulated biogeophysical effect can also be partly attributed to the colder background climate of piControl, with the anthropogenic forcings fixed at the levels of the year 1850.

## The paired forest and openland sites

To obtain the observational evidence on the biogeophysical effect of deforestation on daily temperature variability, the observations from 33 flux sites of the FLUXNET2015 Tier 1 dataset[51] and 7 flux sites of the AmeriFlux dataset[52] are used. These sites are located in North America and Europe, where the deforestation effect on daily temperature variability is prominent, as suggested by the simulations. These sites provide gap-filled observations of daily 2-meter air temperature, 10-meter wind speed, and surface SHF, which are required in this study. The SHF is observed through the eddy-covariance method and has been corrected to address the surface energy imbalance issue.

The land cover types of these 34 sites include forests, grasslands, croplands, and open shrublands. Following the "space-for-time" substitution logic, a forest site is paired with a neighboring openland (grasslands, croplands, and open shrublands) site. To minimize the influence of the elevation difference between the paired forest and openland sites on our analysis, the pairs with elevation differences exceeding 500 m are discarded. In total, 24 pairs are available for further analysis (Supplementary Fig. 5 and Supplementary Table S3). The mean horizontal distance between the paired sites is 24.6 km, and the mean elevation difference is 71.7 m. The paired sites share almost the same background atmospheric conditions due to the short horizontal distance[53]. Therefore, any differences in the near-surface atmosphere (e.g., daily temperature variability) between the paired sites can be mostly attributed to the contrasting land cover types. Again, since the "space-for-time" substitution method is used, only the local effects are considered here, whereas the nonlocal effects are mostly not included. For example, the atmosphere above the paired sites tends to be mixed due to horizontal advection. As a result, the effect related to the change in horizontal near-surface TADV is weakened in this case.

## The Kolmogorov–Smirnov two-sample test

The Kolmogorov–Smirnov two-sample test is a nonparametric test to determine if two samples of data are from the same continuous distribution. The null hypothesis is that the two dataset values are

from the same continuous distribution. The alternative hypothesis is that these two datasets are from different continuous distributions. The hypothesis test is carried out at the confidence level of 95%. This test is performed using MATLAB.

## Data availability

The piControl and historical simulations of the CMIP6, the deforest-globe, hist-noLu and ssp370-ssp126Lu simulations of the LUMIP, the hist-nat simulation of the Detection and Attribution Model Intercomparison Project, and the ssp370 simulation of the Scenario Model Intercomparison Project are available at https://esgf-node.llnl.gov/search/cmip6/. The ERA5 reanalysis data are available at https://www.ecmwf.int/en/forecasts/datasets/reanalysis-datasets/era5. The NCEP-DOE AMIP-II reanalysis data are available at https://psl.noaa.gov/data/gridded/data.ncep.reanalysis2.html. The dataset mapping the biophysical effects of vegetation cover changes is available at https://jeodpp.jrc.ec.europa.eu/ftp/jrc-opendata/ECOCLIM/Biophysical-effects-vgt-change/v2.0/.

## Code availability

All codes of this study are available at Figshare[71].

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

## Acknowledgements

This research is supported by the Natural Science Foundation of China (42130602 awarded to W.G., 42005096 awarded to J.G., 42105023 awarded to B.Z., and 42175136 awarded to B.Q.), Jiangsu Collaborative Innovation Center for Climate Change, and the Frontiers Science Center for Critical Earth Material Cycling of Nanjing University.

## Author contributions

J.G., Q.L., and W.G. conceived and designed the overall study. J.G. and Q.L. performed the data analysis, with help from B.Z., Z.L., S.L., and B.Q. in the interpretation of the results. J.G., Q.L., and W.G. drafted the manuscript. B.Z., Z.L., S.L., and B.Q. edited the manuscript. J.G., Q.L., B.Z., Z.L., S.L., B.Q., and W.G. discussed and revised the manuscript.

## Competing interests

The authors declare no competing interests.
