## [Peer review file · Nature Communications]

Reviewers' comments:

Reviewer #1 (Remarks to the Author):

This paper examines the idealized deforestation simulations from LUMIP within CMIP6 to assess how deforestation impacts temperature variability in different regions of the world and across different seasons. The main result is that temperature variability significantly rises (by 20% or so) in northern mid- to high-latitudes, especially in winter.

Overall, I found the paper to be solid technically. The authors did a good job of evaluating the models and assessing different metrics for temperature variability to establish whether or not the signal is robust. As noted in a comment below, it's unfortunate that one of the key mechanisms identified can only be evaluated with one model, but that is not the fault of the authors. The question I have, then, is whether or not these results are of broad enough impact and relevance for Nature Communications. The response of the authors to the significant points 1 and 2 may affect one's opinion on this. The main question in my mind is whether or not real world deforestation and any potential re/afforestation are likely to generate detectable shifts in temperature variability. This question is not really addressed in this paper, but it is key to understanding whether or not these results are societally-relevant.

Significant points:

1. The use of the idealized deforestation scenario for this analysis makes sense because the large-scale deforestation in that simulation makes it easier to establish signal from noise. However, this did lead me to wonder whether or not real world deforestation over the historical period would have generated a detectable signal as well or not. It would be relatively straightforward to check by looking at the CMIP6 historical simulations vs the hist-noLu simulations in which land use is held fixed at 1850 levels.

2. The above comment leads into this comment. The authors argue at the end of the manuscript that these results imply that the impacts on temperature variability should be considered in the context of reforestation or afforestation activities designed to mitigate emissions. My question is whether or not the re/afforestation that could occur is likely to be of large enough scale to drive a detectable change in temperature variability. The authors argue that small scale changes in forest cover are not sufficient to generate a detectable in temperature variability because the local impacts on fluxes effectively get mixed by the atmosphere. So, is the re/afforestation anticipated with the various Shared Socioeconomic Pathways (SSPs) generated by Integrated Assessment Models sufficient to potentially drive an observable change in temperature variability and also are the anticipated regions of re/afforestation

located in places of high sensitivity of temperature variability to forest cover (e.g., especially in snow-covered northern high latitudes in winter ... which are not ideal locations for re/afforestation anyway due to the warming influence of forests in high latitudes).

3. It's unfortunate that the primary identified mechanism to explain the shift in temperature variability is that associated with horizontal thermal advection, which is a result coming from just one model that archived the fields required to calculate daily timescale thermal advection. This is obviously not the fault of the authors, but it does make it hard to know how much weight to put into this relatively central finding of the study.

--Minor points:

1. Line 15: I don't think it is correct to say that the biophysical impacts of deforestation on temperature variability is unknown. It's more appropriate to say that it is relatively unresearched ... or something like that. We have a pretty good sense at this point what the impacts of deforestation are in a mean sense and across the diurnal cycle.

2. Line 61: "nature-based resolution to mitigate": I think the word solution rather than resolution is more appropriate.

3. Line 88: "DTDTx and DTDTn guarantee the rationality". Guarantee is not appropriate word. Support would be better.

4. Figure S1: You are comparing the daily variability in the PI control for the models against the daily variability from present day in observations. Obviously, there is a mismatch there, which is probably ok, but it should at least be acknowledged.

5. Figure S1: The authors argue that there is good agreement between the model and observations in terms of the DTDTx and DTDTn. How are they defining good agreement. To me, the bias looks pretty significant, especially for winter, which also happens to be the season with the biggest deforestation signal on DTDTx and DTDTn.

6. Lines 272-279: I'm not sure that the discussion of possible biogeochemistry feedback effects is really relevant here. Clearly, deforestation changes the CO₂ concentrations in the atmosphere and that could impact temperature variability, but discussing this here seems kind of a distraction from the main point.

7. The paragraph on local vs large-scale deforestation was a little unclear and would benefit from a careful edit.

Reviewer #2 (Remarks to the Author):

This paper is based on analyses of large-scale deforestation experiments conducted with five global models. The authors show the impacts of broad deforestation on day-to-day temperature variability in each of the four seasons. They conclude that deforestation significantly increases temperature variability in the high latitudes during cold seasons. They attribute this to enhanced meridional thermal advection. They make heavily (and appropriately) caveated assessments of the societal implications of this study in the conclusions.

I do not feel this paper is appropriate for publication in Nature Communications. The analysis, while interesting, has two fundamental problems. First (comment 8 below), the primary mechanism (Figure 4) used to explain the signal (Figure 2) is equally active in all seasons, while the signal is only present in DJF (for daily maximum) or DJF and MAM (for daily minimum). The explanation is not satisfactory or complete. Second (comment 7 below), this study looks to describe variability while choosing to ignore the mean signal. I understand the desire to focus on the question of variability, but one needs to trust that the large-scale behavior in the mean is sensible. If there are flaws in the model behavior completely distorting the mean response, it is likely that these flaws will influence the variability, too. I just don't think you can ignore it completely. For these reasons, I do not feel the paper should be published in Nature Communications.

Specific comments:

1. Introduction: sound rationale motivating the study, but it doesn't rise to Nature Communications level in my mind.
2. Fourteen supplemental figures is way too many! Good rule of thumb: don't have more figures in the supplement than you have in the main text. There's way too much buried here. Here are some suggestions:
 - a. Figures S1 and S2 comparing DTDTx and DTDTn for Reanalysis (1981-2010) with 30 years of the piControl run are jam-packed with interesting information! They also duplicate two full columns. Why not make S1 DTDTx for the three computations (HadGHCND, NCEP/NCAR, MMM) and S2 DTDTn for the three. That will make it easier for the reader to directly compare the two historical products with each other.
 - b. Figures S4-S7 seem entirely consistent with the earlier figures and analyses. I don't think they add value. You can mention that you get similar results with this different metric of variability.
 - c. Figure S8-S10 can be replaced with a sentence or two after Figure 3 stating if the PDF changes are statistically significant or not (as mentioned in the comment about Figure 3).
 - d. Figure S11 seems like a pretty critical figure. I don't think you can avoid this piece of the story! Figure S12 supports Figure S11, but is slightly less crucial.
3. Lines 105-6: This claim that DTDTx and DTDTn both increase in three seasons in the northern mid- to high-latitudes does not seem consistent with Figure 2. It seems correct for DTDTn, but DTDTx only shows broad increases in DJF. In MAM there is very little coherent signal north of 50N (and what there is

is blue). In SON, there is no high-latitude signal in North America or Asia, and only a small positive signal in eastern Europe. This picture is in conflict with the claim of “all models” agreeing (lines 109-112).

4. Figure 3 (and associated text): Have you verified the statistical significance of the differences between the PDFs shown in Figure 3? Can you quantify the percentage of the distribution that moves from the near-zero region to the tail region? My visual estimate is 3-4% of the distribution gets shifted from (-3,3) to the tails beyond this region. That may be statistically significant. If you can perform such tests on these pdfs, you can probably remove Figures S8-S10 and instead just mention if the PDF changes in each season are significant or not.

5. Lines 198-200: You need to include descriptions of how you calculate each of these three terms. (Later I got to the methods and saw that the equations are provided in the methods: you need to mention this in the main text.) You need to convince the reader that these effects are easy to separate.

6. Lines 204-6: You claim that Figure 4 shows that this mechanism largely explains the signal in Figure 2. But Figure 2 only shows a response in some seasons; Figure 4 shows nearly equal effects in all seasons. This seems to contradict your rationale. Your physical explanation associated with Figure 5 (through line 220) supports your claim, but the fact that this mechanism is equally active in the summer (when there is no corresponding signal in Figure 2) makes it difficult to believe that you’ve captured the full story.

7. Line 286: I’m troubled by “we do not question the rationality of the simulated cooling effects of deforestation.” Shouldn’t you always question the rationality of the mean behavior of the primary variable you are investigating? As I said with regard to Figure S11, I don’t think you can avoid this part of the story!

8. I expected Figure S13 to be in equivalent units as Figure 4 since you describe them both as “the effect of ...”. In the text you seem to downplay the sensible heat flux effect because it is strongest in JJA and SON. You should present these two figures in comparable ways so we can see if perhaps the summer-time effect in Figure S13 counters the thermal advection affect shown in all seasons in Figure 4. That might be the answer to the fundamental unanswered question I have about this paper!

Reviewer #3 (Remarks to the Author):

Review of "Deforestation increases the daily temperature variability in cold seasons and in the northern mid- to high-latitudes" by Ge *et al.*

In their study, Ge *et al.* analyze the effect of deforestation on the day-to-day (DTD) variations in surface air temperature for different seasons using Earth system model (ESM) simulations from the Land Use Model Intercomparison Project LUMIP, part of CMIP6. The authors find that the DTD variability increases for daily maximum and minimum temperature, but the changes are only noteworthy during the winter months in North America and Eurasia. By decomposing a thermodynamic equation using the model output of one ESM, they disentangle the different effects contributing to the increased variability. They find that increased horizontal advection of warm air is the main cause for increased DTD variability rooted in increased meridional pressure and temperature gradients.

While the study and its results are interesting, the drawn conclusions are overstated. Please see my General Comments for details. Also, the model simulations used in this study are quite idealized, mainly because all other historical drivers that impact on DTD variability are switched off in the model setup. The found effect might vanish when the other drivers are considered. This aspect needs to be addressed, because the narrative of the manuscript stresses the importance of this effect and "its broad impacts on public health, ecosystem functions and the economy". Overall, the structure of the manuscript is clear and concise, the English language, however, needs some proofreading, especially towards logic linking of sentences.

Bottom line, the study is definitely interesting, but the findings are not strong and significant enough to back the conclusions drawn. In its current state, I cannot recommend the manuscript for publication in *Nature Communications*. I, however, do recommend and encourage publication in a more specialized community journal after some revisions. Please see my comments below.

1 General Comments:

1.1 *I wonder whether the signals you find are really noteworthy. First, the strength of the deforestation effect on the DTD variability should be compared to the effect strength of other drivers, such as rising CO₂. This would help to get an idea whether this effect is significant enough and actually provides enough support to your conclusions, e.g., the statement, that the effects of forest changes on daily temperature variability need to be considered when implementing large-scale afforestation to mitigate climate change. Further, in your Figures S1 and S2 you present a comparison of your variability metrics between observations/reanalysis and the multi-model mean. Based on this results you argue "Validated against the observational and the reanalysis data, the five models can well reproduce the spatial and temporal variations... ." And further "The reasonable simulations ... guarantee the rationality of further studies based on the models." I am not convinced that the models reproduce the reanalysis and observations well, especially in the regions and seasons where you find the strongest signal, so cold season and high-mid latitudes. In comparison to HadGHCND there is a bias of at least 3° C in the models for DJF*

and North America / Eurasia. Comparing to reanalysis, the bias is very strong in SON, where models underestimate DTD variability by a few degrees. The largest change due to deforestation you find is around 0.5 °C. Given the high uncertainty, the signal due to deforestation might be over-interpreted and conclusion far-fetched.

1.2 I do not see that the increase in day-to-day temperature variability due to deforestation in the cold season in the mid- to high-latitudes has a "broad effect on public health, ecosystem functions and the economy" (L24). However, this is the central conclusion of your study. The strongest increase in DTD variability is around 0.5 °C in winter months according to your Figure 2. The ecosystems functions are likely not affected by these small changes, especially because most of the ecosystems there are in their dormant phase in winter. Also economic growth is likely not being affected by this small increase in DTD variability. You cite Kotz et al. (2021) here, who actually write that there is no effect of winter DTD variability on economic growth rates. Regarding public health you state "The reduced daily temperature variability in cold seasons is commonly beneficial to public health as people are in particular susceptible to a variety of disease in these seasons" and you cite the reference which is listed under number 39. Reference 39 is the LUMIP model experiment design paper. I doubt that an increase of 0.5 °C in DTD variability in winter has a measurable effect on human health – in summer maybe, but not in winter.

1.3 I find the decomposition using thermodynamic equation an interesting and important analysis. Can you elaborate a bit on where the effect of surface roughness changes and changes in evaporative cooling enter the equation? Are these processes already covered by the first and third term, so by looking at horizontal advection and the estimates of surface sensible heat fluxes? Is it really enough to look into surface sensible heat fluxes when studying the diabatic heating term? Deforestation is associated with concomitant changes in surface albedo and surface latent heat fluxes - should that not be accounted for in your analysis? Some studies also show that deforestation can influence low-level cloud cover impacting on the amount of incoming short-wave and outgoing long-wave radiation. This again can modulate day-to-day variability of the surface air temperature. Could this cloud feedback play a role here, too, as proposed by Duveiller et al. (2021)? The decomposition is only done for one model (CanESM5). The dynamics between the different GCMs + land-surface models is quite pronounced, so are the results reliable if they are only based on one model? And if so, why would CanESM5 be a good model to study this decomposition besides the fact that it is the only model for which the necessary output is accessible.

2 Specific Comments:

2.1 L14: What about surface roughness changes due to deforestation?

2.2 L17: 'Show' is a strange word here. I would rather write: 'Here, we investigate / analyze ... '

2.3 L18: There is only 'the global scale', not multiple global scales.

2.4 LL30-31: Leaf area index and surface albedo is related, however, evapotranspiration should be mentioned here.

2.5 LL43-45: That does not make logical sense. Why "but some studies, in contrast..."? Where is the contrast?

2.6 L47: What is a 'temperature series'? You mean 'time series' of temperature, right?

2.7 LL84-85: *In the literature I mostly see T_{\max} and T_{\min} as abbreviations. And what about daily mean temperatures?*

2.8 L118: *I would not say that these are different results. You just present the same results in a different way (absolute versus relative changes).*

2.9 *Non-local effects are captured by the observations - we just cannot disentangle them from local effects or other drivers such as the greenhouse gas effect, etc.*

2.10 LL399: *Please provide a list of the variable names you used from the CMIP output in the section 'Attributions of changes in the daily temperature variability'. This is important so that others can easily reproduce your work.*

2.11 L403: *Can you provide a reference of the equation used.*

2.12 *Figures: Please add a unit label to the color bar! For Figure 5, also a legend arrow for wind speed is missing. In the current version, the reader cannot assess how strong the changes in wind speed actually are.*

References

Duveiller, G., Filipponi, F., Ceglar, A., Bojanowski, J., Alkama, R., and Cescatti, A. (2021). Revealing the widespread potential of forests to increase low level cloud cover. *Nature Communications*, 12(1):4337.

Response to the comments

Reviewer #1 (Remarks to the Author):

This paper examines the idealized deforestation simulations from LUMIP within CMIP6 to assess how deforestation impacts temperature variability in different regions of the world and across different seasons. The main result is that temperature variability significantly rises (by 20% or so) in northern mid- to high-latitudes, especially in winter.

Overall, I found the paper to be solid technically. The authors did a good job of evaluating the models and assessing different metrics for temperature variability to establish whether or not the signal is robust.

Response: Thank you.

As noted in a comment below, it's unfortunate that one of the key mechanisms identified can only be evaluated with one model, but that is not the fault of the authors. The question I have, then, is whether or not these results are of broad enough impact and relevance for Nature Communications.

Response: We provide more evidence from both simulations and observations to support the key mechanisms in the revised manuscript. Please see the detailed response to the significant point 3 below.

The response of the authors to the significant points 1 and 2 may affect one's opinion on this. The main question in my mind is whether or not real world deforestation and any potential re/afforestation are likely to generate detectable shifts in temperature variability. This question is not really addressed in this paper, but it is key to understanding whether or not these results are societally-relevant.

Response: We have carefully addressed the significant points 1 and 2 following the suggestion of the reviewer. Please see the detailed responses to the significant points 1

and 2 below.

Significant points:

1. The use of the idealized deforestation scenario for this analysis makes sense because the large-scale deforestation in that simulation makes it easier to establish signal from noise. However, this did lead me to wonder whether or not real world deforestation over the historical period would have generated a detectable signal as well or not. It would be relatively straightforward to check by looking at the CMIP6 historical simulations vs the hist-noLu simulations in which land use is held fixed at 1850 levels.

Response: We thank the reviewer for the contrastive comment. Following the suggestion of the reviewer, we investigate the biogeophysical effects of deforestation during the historical period (1850-2014) with the *historical* simulation from the CMIP6 and the *hist-noLu* simulation from the LUMIP.

It should be emphasized that there are two issues when the effect of historical deforestation is examined. First, during the historical period, the tree cover changes are heterogeneous in space, with the concurrence of both forest losses and gains. This may lead to the effect of forest losses being offset by the effect of forest gains as well as the effects of other land use and land management. Second, the models agree less on the tree cover changes during the historical period, although the historical land cover changes are prescribed by the same dataset.

Nevertheless, we find large-scale deforestation in North America during the historical period. Such deforestation is spatially coherent and has higher intermodel consistency. Thus, we take North America as a case study. We find detectable increases in daily temperature variability due to historical deforestation in North America. This result is highly consistent with the result obtained from the idealized deforestation scenario.

Please see Fig. 5 and the revised manuscript:

Lines 307-345: The effects of historical deforestation on daily temperature variability. The above analysis is built on the idealized deforestation scenario, which is unlikely to occur in the real world. We therefore wonder whether the real-world deforestation over the historical period would have generated a detectable signal as well. To this end, the historical simulation (*historical*) of the CMIP6 and the historical no land-use simulation (*hist-noLu*) of the LUMIP are used. The *historical* covers the period of 1850–2014 with evolving and externally imposed natural and anthropogenic forcings (e.g., GHGs, aerosols and LULCCs; see “Methods”). The *hist-noLu* is identical to the *historical*, except that LULCCs are fixed at preindustrial levels (see “Methods”). The last 30-year simulations (1985–2014) of the *historical* and *hist-noLu* are compared (*hist-noLu* minus *historical*) to isolate the deforestation effect. Nine earth system models are used here given the model output availability (Table S1). These models are also reasonable in the DTD simulations validated against the reanalysis (Figs. S1e-1h and S1i-1p; see “Methods”).

Tree cover changes during the historical period are highly heterogeneous in space with the concurrence of forest losses and gains (Fig. S7), which may confound the signals arising from forest losses and gains. Exceptionally, we find tremendous and spatially coherent tree cover losses in North America since 1850 (Fig. 5a). Moreover, this decreasing trend of tree cover is also consistent across the models, with the multimodel mean value of the net tree cover loss being approximately -1.3 million km² during the historical period of 1850-2014 (Fig. 5b). As a result of such large-scale deforestation, most models simulate increases in wintertime DTD (with multimodel mean values of 0.2 - 0.5 °C or 2 - 12%; Fig. 5c) and SDT (with multimodel mean values of 0.2 - 0.6 °C or 2 - 10%; not shown). In other words, historical deforestation has led to a detectable increase in the wintertime daily temperature variability in North America. The enhanced daily temperature variability can be attributed to the higher TADV (Fig. 5e) and simultaneously is slightly offset by the lower DTDSHF (Fig. 5d). These mechanisms are consistent with those for the deforestation effects shown in the idealized deforestation scenario. Unfortunately, again only one model (ACCESS-

ESM1-5) supports the analysis of the TADV response to the historical deforestation here due to the data availability. Nevertheless, the wintertime TADV response to deforestation shown by the ACCESS-ESM1-5 model (Fig. 5e) is broadly consistent with the CanESM5 model (Fig. 3e). More importantly, the consistent DTDT or SDT responses to the idealized deforestation scenario and the historical deforestation scenario as well as the consistent mechanisms consolidate the robustness of the results.

Note that there is some spatial mismatch between the DTDT change (Fig. 5c) and the TADV change (Fig. 5e). This mismatch is because the pattern of tree cover changes for the ACCESS-ESM1-5 model (Fig. S7a) is slightly different from the multimodel mean pattern (Fig. 5a).

Lines 500-533: **Investigation of the biogeophysical effect of historical deforestation.**

To obtain the biogeophysical effects of deforestation during the historical period, the historical simulation (*historical*) of the CMIP6⁴³ and the historical no land-use simulation (*hist-noLu*) of the LUMIP⁴⁴ are used.

The *historical* is a fully coupled simulation covering the historical period of 1850-2014. This simulation is branched from the *piControl* and is driven by evolving and externally imposed forcings. All natural (e.g., volcanic aerosols and solar variability) and anthropogenic (e.g., GHGs, aerosols and LULCCs) forcings are included and are largely based on observations. The historical LULCCs are prescribed by the new generation of the Land-Use Harmonization 2 (LUH2) dataset⁶¹ and include detailed changes in land cover, land use and land management.

The *hist-noLu* is the same as the *historical* except with LULCCs fixed at the levels of year 1850. Note that the atmospheric CO₂ concentration in the *hist-noLu* is identical to the value in the *historical*. Thus, the biogeophysical effect of historical deforestation is isolated by comparing the last 30-year (1985-2014) simulations of the *historical* and *hist-noLu* (*historical* minus *hist-noLu*). By the time of this study, nine earth system models contribute to both the *historical* and *hist-noLu* simulations and provide the

required daily 2-meter temperature output; thus, these nine models are used here (Table S1). For each model, multiple members are available for the *historical* and *hist-noLu*. To save computational cost, the first member of the *historical* and *hist-noLu* are used for analysis.

The total tree cover change during the historical period is obtained by subtracting the last 30-year (1985-2014) mean tree cover fraction of the *hist-noLu* from the corresponding value of the *deforest-globe*. Note that while the historical LULCCs are prescribed by the LUH2, the implementation of the historic LULCCs across different models is commonly different (Fig. S7), and this problem has always plagued model intercomparison⁴⁴. Such intermodel consistency may cause the models to diverge in the response of daily temperature variability to the historical LULCCs. Moreover, both tree losses and gains may occur in a region, confounding the signals in daily temperature variability arising from forest losses and gains. Fortunately, we find a large area of forest losses during the historical period in North America (Fig. 5a). This tree cover change is considerable, spatially coherent and of high intermodel consistency. Therefore, we take North America as a case study to investigate the response of daily temperature variability to historical deforestation.

The above comment leads into this comment. The authors argue at the end of the manuscript that these results imply that the impacts on temperature variability should be considered in the context of reforestation or afforestation activities designed to mitigate emissions. My question is whether or not the re/afforestation that could occur is likely to be of large enough scale to drive a detectable change in temperature variability. The authors argue that small scale changes in forest cover are not sufficient to generate a detectable in temperature variability because the local impacts on fluxes effectively get mixed by the atmosphere. So, is the re/afforestation anticipated with the various Shared Socioeconomic Pathways (SSPs) generated by Integrated Assessment Models sufficient to potentially drive an observable change in temperature variability and also are the anticipated regions of re/afforestation located in places of high

sensitivity of temperature variability to forest cover (e.g., especially in snow-covered northern high latitudes in winter ... which are not ideal locations for re/afforestation anyway due to the warming influence of forests in high latitudes).

Response: This is also a constructive comment. Following the suggestion of the reviewer, we investigate the effects of potential afforestation or reforestation anticipated by the various shared socioeconomic pathways (SSPs) on daily temperature variability. To this end, the *ssp370* simulation from the ScenarioMIP and the *ssp370-ssp126Lu* from the LUMIP are used. The *ssp370* covers the future period of 2014–2100 under the SSP3-7.0 scenario with substantial deforestation. The *ssp370-ssp126Lu* is the same as the *ssp370* except that land use is from the SSP1-2.6 scenario (afforestation scenario). Accordingly, the effect of afforestation anticipated by the SSP1-2.6 scenario can be obtained by comparing the *ssp370-ssp126Lu* and *ssp370* simulations (*ssp370-ssp126Lu* minus *ssp370*).

Please see Fig. 6 and the revised manuscript:

Lines 379-403: **The effects of potential afforestation on daily temperature variability.** Based on the above results, we speculate that afforestation in the northern extratropics may reduce daily temperature variability. To test this speculation, the scenario simulation (*ssp370*) of the Scenario Model Intercomparison Project⁵⁶ (ScenarioMIP) and the future land-use policy sensitivity simulation (*ssp370-ssp126Lu*) of the LUMIP are used. The *ssp370* covers the future period of 2014–2100 under the shared socioeconomic pathway (SSP) 3-7.0 scenario (SSP3-7.0) with substantial deforestation and high emissions (see “Methods”). The *ssp370-ssp126Lu* is identical to the *ssp370* except that LULCCs are from the SSP1-2.6 scenario with substantial afforestation or reforestation. The last 30-year simulations (2071–2100) of the *ssp370* and *ssp370-ssp126Lu* are compared (*ssp370-ssp126Lu* minus *ssp370*) to isolate the afforestation effect. Six earth system models are used here given the model output availability (Table S1).

Fig. 6a shows the differences in tree cover between the SSP1-2.6 and SSP3-7.0 scenarios by the end of the 21st century. Benefiting from afforestation or reforestation anticipated by the SSP1-2.6 scenario, we can see increases in tree cover in some northern extratropical regions, e.g., eastern America, Europe and eastern China. Figs. 6b-6e show the biogeophysical effects of these tree cover changes on DTDT for each season. As expected, the models consistently project that the local and regional DTDT will be reduced by 0.09-0.21 °C in winter, spring and summer due to 9-21% increases in tree cover fractions in eastern America by the end of the 21st century. For Europe, however, the afforestation effects on DTDT are mostly small, probably due to the small increase in tree cover anticipated by the SSP1-2.6 scenario (mostly within 12%). While the tree cover changes in eastern China are considerable, the effects on DTDT are also limited throughout the year. This small DTDT response can be attributed to the low sensitivity of DTDT to forest changes in China (Figs. 1c-1f).

Lines 547-569: **Investigation of the biogeophysical effects of potential afforestation.** To obtain the biogeophysical effects of potential afforestation by the end of the 21st century, the *ssp370* simulation of the Scenario Model Intercomparison Project⁵⁶ (ScenarioMIP) and the *ssp370-ssp126Lu* simulation of the LUMIP are used.

The *ssp370* is a fully coupled simulation covering the future period of 2014-2100 under the SSP3-7.0 scenario. This scenario represents the medium to high end of the range of future forcing pathways with substantial LULCCs (in particular global forest losses) and high emissions of near-term climate forcers (namely, tropospheric aerosols, tropospheric O₃ precursors, and CH₄). The *ssp370-ssp126Lu* is the same as the *ssp370*, but LULCCs are from the SSP1-2.6 scenario. The SSP1-2.6 scenario represents the low end of the range of future forcing pathways that aim to constrain the global mean temperature below 2 °C by 2100. In contrast to the SSP3-7.0 scenario, there are substantial forest gains due to afforestation or reforestation for the SSP1-2.6 scenario. The LULCCs for both the SSP1-2.6 and SSP3-7.0 are prescribed by the LUH2 dataset. Note that the atmospheric CO₂ concentration in the *ssp370* is identical to the value in

the *ssp370-ssp126Lu*. Thus, the biogeophysical effect of potential afforestation is isolated by comparing the last 30-year (2071-2100) simulations of the *ssp370* and *ssp370-ssp126Lu* (*ssp370-ssp126Lu* minus *ssp370*).

By the time of this study, six earth system models contribute to both the *ssp370* and *ssp370-ssp126Lu* simulations and provide the required daily 2-meter temperature output; thus, these four models are used here (Table S1). For each model, only one member is available for the *ssp370-ssp126Lu*. Thus, the single member of the *ssp370-ssp126Lu* and the corresponding member of the *ssp370* are used for analysis.

2. It's unfortunate that the primary identified mechanism to explain the shift in temperature variability is that associated with horizontal thermal advection, which is a result coming from just one model that archived the fields required to calculate daily timescale thermal advection. This is obviously not the fault of the authors, but it does make it hard to know how much weight to put into this relatively central finding of the study.

Response: We acknowledge the limitation that only a single model is used to analyze the mechanisms in the original manuscript. To address this issue, we provide more evidence from observations and simulations to support the DTDT signal and the mechanisms in the revised manuscript.

First, we analyze the mechanisms for the effect of historical deforestation. Unfortunately, again, only a model (ACCESS-ESM1-5) can support the analysis of horizontal temperature advection. Nevertheless, the response of the horizontal temperature advection to historical deforestation shown by ACCESS-ESM1-5 is fully consistent with the response to idealized deforestation shown by CanESM5.

Please see Fig. 5 and the revised manuscript:

Lines 331-341: The enhanced daily temperature variability can be attributed to the higher TADV (Fig. 5e) and simultaneously is slightly offset by the lower DTDSHF (Fig.

5d). These mechanisms are consistent with those for the deforestation effects shown in the idealized deforestation scenario. Unfortunately, again only one model (ACCESS-ESM1-5) supports the analysis of the TADV response to the historical deforestation here due to the data availability. Nevertheless, the wintertime TADV response to deforestation shown by the ACCESS-ESM1-5 model (Fig. 5e) is broadly consistent with the CanESM5 model (Fig. 3e). More importantly, the consistent DTDT or SDT responses to the idealized deforestation scenario and the historical deforestation scenario as well as the consistent mechanisms consolidate the robustness of the results.

Second, an increase in horizontal temperature advection can be caused by an increase in wind speed and/or an increase in horizontal temperature gradient. We therefore analyze the responses of wind speed and temperature gradient to deforestation based on all models. Here, we calculate the wind speed and horizontal temperature gradient based on monthly 10-meter wind and 2-meter temperature, which are output by all models. The models consistently suggest increases in both wind speed and horizontal temperature gradient, favoring the increase in horizontal temperature advection.

Please see Fig. 4 and the revised manuscript:

Lines 287-298: While the result on the TADV response to deforestation is from a single model, this result is also indirectly supported by other models and the observations from the paired sites. An increase in TADV can be caused by an increase in near-surface wind speed or horizontal temperature gradient (see Methods). Fig. 4 shows the responses of the near-surface wind speed and horizontal temperature gradient to deforestation for each season. All the models (Figs. 4a-4d) and the in situ observations (Figs. S5e) consistently suggest an increase in near-surface horizontal wind speed due to the lower surface roughness in deforested areas or the openland sites, which tends to increase TADV. The models also consistently suggest an enhanced near-surface horizontal temperature gradient in some midlatitude deforested areas in winter and spring (Figs. 4e-4h), also favoring the increase in TADV. This enhanced horizontal temperature gradient can be explained by the strong cooling effect of deforestation in boreal regions

(e.g., Fig. S3).

Third, we use 24 pairs of spatially adjacent forest and openland sites in North America and Europe. We also find that daily temperature variability is overall higher over the openland sites than the neighboring forest sites in winter. This result is consistent with the simulations. Moreover, the observations indicate a larger wind speed over the openland sites than the neighboring forest sites, favoring a higher horizontal temperature gradient over the openland sites.

Please see Fig. S5 and the revised manuscript:

Lines 166-178: In addition to the simulations, we attempt to find observational evidence on the deforestation effects on daily temperature variability. To this end, 24 selected pairs of spatially adjacent forest and openland sites in North America and Europe are used⁵¹⁻⁵³ (Fig. S5a and 5b; Table S3; see “Methods”). Fig. S5c shows the box plots of the DTDT or SDT differences between the paired forest and openland sites (openland minus forest) for each season. Both the DTDT and SDT differences are positively biased in winter, implying overall higher daily temperature variability over the openland sites than the neighboring forest sites. This result confirms the enhanced wintertime daily temperature variability in the northern extratropical deforested areas in the simulations. For other seasons, however, the DTDT and SDT differences are either negatively biased or unbiased. This result implies that the DTDT or SDT of the openland sites are overall smaller than or identical to the values of the neighboring forest sites. This result is in contrast with the simulated result, which is discussed later.

Lines 245-247: The DTDSHF response to deforestation is also supported by the observational evidence from the paired sites in North America and Europe (Figs. S5d).

Lines 292-294: All the models (Figs. 4a-4d) and the in situ observations (Figs. S5e) consistently suggest an increase in near-surface horizontal wind speed due to the lower surface roughness in deforested areas or the openland sites, which tends to increase

TADV.

Last, we also acknowledge the uncertainty in the mechanisms and give a discussion on this point.

Lines 444-454: We acknowledge the uncertainty in the mechanisms for the biogeophysical effects of deforestation on daily temperature due to the model output availability. In particular, the effect related to horizontal near-surface temperature advection is examined based on limited models. It should be emphasized that, despite this limitation, the deforestation effects on daily temperature variability are robust because the models and observations consistently agree on these effects, and the results are independent of the metrics of the daily temperature variability. Since the daily variables are required to study the daily temperature variability, we encourage more model groups that contribute to the CMIP6 and LUMIP to report daily output to support further analysis. We also encourage other groups to study the mechanisms for the deforestation effects on daily temperature variability with their own models and experiments.

Minor points:

1. Line 15: I don't think it is correct to say that the biophysical impacts of deforestation on temperature variability is unknown. It's more appropriate to say that it is relatively unresearched ... or something like that. We have a pretty good sense at this point what the impacts of deforestation are in a mean sense and across the diurnal cycle.

Response: The reviewer may slightly misunderstand the temperature variability because temperature variability is commonly confused with temperature extremes. While an increase in variability leads to increases in the frequency and magnitude of extremes, they are not the same. Here, temperature variability reflects changes in temperature between neighboring days. We agree with the reviewer that "*We have a*

pretty good sense at this point what the impacts of deforestation are in a mean sense and across the diurnal cycle” because the deforestation effects on the mean temperature, the diurnal cycle of temperature and temperature extremes are indeed broadly documented. However, as far as we know, the deforestation effects on temperature variability (namely, changes in temperature between neighboring days) remain largely unknown.

To avoid ambiguity, we further differentiate temperature variability from the diurnal cycle of temperature or temperature extremes in the introduction section in the revised manuscript:

Lines 51-54: Although the deforestation effects on the mean, the diurnal cycle and extremes of temperature have been widely documented, how deforestation influences temperature variability remains largely unknown. Temperature variability describes fluctuations in a time series of temperature at various (from daily to decadal) time scales.

Lines 60-65: Statistically speaking, an increase in variability commonly coincides with a higher tail probability, implying more frequent weather and climate extreme events³⁰,³¹ and consequently widespread adverse impacts on nature and humans³². Therefore, understanding the deforestation effects on temperature variability is as important as understanding the deforestation effects on the temperature mean and extremes.

2. Line 61: “nature-based resolution to mitigate”: I think the word solution rather than resolution is more appropriate.

Response: We change “resolution” to “solution” as suggested by the reviewer in the revised manuscript (please see line 74).

3. Line 88: “DTDTx and DTDTn guarantee the rationality”. Guarantee is not appropriate word. Support would be better.

Response: We delete this sentence due to other required revisions in the revised

manuscript.

4. Figure S1: You are comparing the daily variability in the PI control for the models against the daily variability from present day in observations. Obviously, there is a mismatch there, which is probably ok, but it should at least be acknowledged.

Response: As suggested by the reviewer, we emphasize this mismatch in the revised manuscript:

Lines 666-669: Note that the anthropogenic forcings for the *piControl* are fixed at the levels of year 1850, which are inconsistent with the reanalysis levels. Thus, there might be larger biases in DTDt and SDT simulated by the *piControl* than the *historical*.

5. Figure S1: The authors argue that there is good agreement between the model and observations in terms of the DTDt_x and DTDt_n. How are they defining good agreement? To me, the bias looks pretty significant, especially for winter, which also happens to be the season with the biggest deforestation signal on DTDt_x and DTDt_n.

Response: In the original manuscript, we use the HadGHCND dataset and NCEP-NCAR reanalysis as the observational benchmarks. However, we find that the use of these two datasets might be problematic. The HadGHCND dataset was produced based on station observations and spatial interpolation. We find considerable missing data in some grids for the HadGHCND dataset, probably leading to a larger uncertainty in this dataset. The NCEP-NCAR reanalysis was produced in the 1990s and is out of date. In the revised manuscript, we replace the HadGHCND dataset and NCEP-NCAR reanalysis with the ERA5 and NCEP-DOE AMIP-II reanalysis datasets. The ERA5 is the state-of-the-art reanalysis product from ECMWF. The NCEP-DOE AMIP-II is the new generation of the NCEP-NCAR reanalysis. The ERA5 and NCEP-DOE AMIP-II reanalysis datasets have also been used to study daily temperature variability in recent studies (e.g., Blackpot et al. 2021).

Please see Fig. S1, S3 and the revised manuscript:

Lines 121-128: The model performance in the DTDT simulation is first evaluated with two dependent reanalysis datasets: the ERA5 reanalysis⁴⁶ and the NCEP-DOE AMIP-II reanalysis⁴⁷ (see “Methods”). Validated with the reanalysis, the models can reasonably simulate the spatial pattern of DTDT with larger values at higher-latitude continents (Figs. S1a-1d and S1i-1p). Moreover, DTDT is overall larger in winter (December, January and February), followed by spring (March, April and May) and autumn (September, October and November), and lowest in summer (June, July and August) in the northern extratropics.

Lines 160-161: The model behavior in the SDT simulation is also reasonable when validated against the reanalysis (Figs. S3a-3d and S3i-3p).

Lines 650-678: **Validation of the model simulation of daily temperature variability.**

To validate the model behavior in the simulations of DTDT and SDT, two dependent reanalysis datasets are used: the ERA5 reanalysis⁴⁶ from the European Center for Medium Weather Forecasting and the NCEP-DOE AMIP-II reanalysis⁴⁷ from the National Centers for Environmental Prediction (NCEP) and National Center for Atmospheric Research (NCAR). These two reanalysis datasets can provide reliable observations of daily temperature variability³⁵ and are widely used in the previous studies^{35, 37}. The ERA5 reanalysis provides the global daily mean 2-meter temperature at a horizontal resolution of 0.25° (latitude) \times 0.25° (longitude) since 1950. The NCEP-DOE AMI-II reanalysis provides global daily mean 2-meter temperature on the T62 Gaussian grid (192 longitude grids \times 94 latitude grids) since 1979.

The 30-year (1985–2014) mean DTDT and SDT from the ERA5 reanalysis (Figs. S1i-1l and S3i-3l) and the NCEP-DOE AMIP-II reanalysis (Figs. S1m-1p and S3m-3p) are used as the observational benchmarks. The 30-year (year 51–80) mean DTDT and SDT simulated by the *piControl* (Figs. S1a-1d and S3a-3d) and the 30-year mean (1985–2014) DTDT and SDT simulated by the *historical* (Figs. S1e-1h and S3e-3h) are

validated against the reanalysis. Note that the anthropogenic forcings for the *piControl* are fixed at the levels of year 1850, which are inconsistent with the reanalysis levels. Thus, there might be larger biases in DTDT and SDT simulated by the *piControl* than the *historical*.

Validated with the reanalysis, both the *piControl* and *historical* can reasonably simulate the global patterns of DTDT and SDT for each season. Statistically speaking, compared to the ERA5 reanalysis, the root mean squared error (RMSE) in the simulated DTDT is between 0.17–0.35 °C for the *piControl* and between 0.16–0.24 °C for the *historical* across the seasons. When compared to the NCEP-DOE AMIP-II reanalysis, the RMSE in DTDT is slightly larger for both the *piControl* (0.25–0.37 °C across the seasons) and *historical* (0.23–0.47 °C across the seasons). In general, the RMSE in the simulated DTDT is highest in winter, followed by spring and autumn, and lowest in summer. A similar result can be obtained for the simulated SDT.

6. Lines 272-279: I'm not sure that the discussion of possible biogeochemistry feedback effects is really relevant here. Clearly, deforestation changes the CO₂ concentrations in the atmosphere and that could impact temperature variability, but discussing this here seems kind of a distraction from the main point.

Response: We agree with this comment, and we delete the discussion on the biochemical effects as suggested by the reviewer in the revised manuscript.

7. The paragraph on local vs large-scale deforestation was a little unclear and would benefit from a careful edit.

Response: We delete this paragraph due to other required revisions in the revised manuscript.

Reviewer #2 (Remarks to the Author):

This paper is based on analyses of large-scale deforestation experiments conducted with five global models. The authors show the impacts of broad deforestation on day-to-day temperature variability in each of the four seasons. They conclude that deforestation significantly increases temperature variability in the high latitudes during cold seasons. They attribute this to enhanced meridional thermal advection. They make heavily (and appropriately) caveated assessments of the societal implications of this study in the conclusions.

I do not feel this paper is appropriate for publication in Nature Communications. The analysis, while interesting, has two fundamental problems. First (comment 8 below), the primary mechanism (Figure 4) used to explain the signal (Figure 2) is equally active in all seasons, while the signal is only present in DJF (for daily maximum) or DJF and MAM (for daily minimum). The explanation is not satisfactory or complete.

Response: We have updated the mechanisms in the revised manuscript. The current mechanisms can well explain the seasonal variation in the signal. Please see the detailed response to the specific comment 6 below.

Second (comment 7 below), this study looks to describe variability while choosing to ignore the mean signal. I understand the desire to focus on the question of variability, but one needs to trust that the large-scale behavior in the mean is sensible. If there are flaws in the model behavior completely distorting the mean response, it is likely that these flaws will influence the variability, too. I just don't think you can ignore it completely. For these reasons, I do not feel the paper should be published in Nature Communications.

Response: We accept this criticism. Following the suggestion of the reviewer, we evaluate the model representation of the deforestation effects on mean temperature with satellite observations in the revised manuscript. Please see the detailed response to the

specific comment 7 below.

Specific comments:

1. Introduction: sound rationale motivating the study, but it doesn't rise to Nature Communications level in my mind.

Response: We have rewritten the introduction section to highlight the importance and significance of this study, for example:

Lines 51-53: Although the deforestation effects on the mean, the diurnal cycle and extremes of temperature have been widely documented, how deforestation influences temperature variability remains largely unknown.

Lines 63-65: Therefore, understanding the deforestation effects on temperature variability is as important as understanding the deforestation effects on the temperature mean and extremes.

Lines 68-73: Changes in daily temperature variability were previously attributed to anthropogenic greenhouse gases (GHGs) emissions³⁴⁻³⁷, aerosols³⁵⁻³⁷, urbanization³⁹ and internal climate variability^{36, 37} but were rarely linked to forest changes. Given the large-scale deforestation worldwide during the historical period^{1, 2}, examining the deforestation effects on daily temperature variability can improve our understanding of human contributions to the evolution of daily temperature variability.

Lines 78-80: Therefore, understanding the afforestation effects on daily temperature variability can help to avoid unanticipated climatic consequences following the implementation of large-scale afforestation.

Lines 94-98: This study fills the knowledge gap regarding the deforestation effects on temperature variability. On the other hand, this study provides new insight into human influences on daily temperature variability through deforestation or afforestation, which

has implications for policy decisions on implementing large-scale afforestation.

2. Fourteen supplemental figures is way too many! Good rule of thumb: don't have more figures in the supplement than you have in the main text. There's way too much buried here. Here are some suggestions: a. Figures S1 and S2 comparing DTDTx and DTDTn for Reanalysis (1981-2010) with 30 years of the piControl run are jam-packed with interesting information! They also duplicate two full columns. Why not make S1 DTDTx for the three computations (HadGHCND, NCEP/NCAR, MMM) and S2 DTDTn for the three. That will make it easier for the reader to directly compare the two historical products with each other. b. Figures S4-S7 seem entirely consistent with the earlier figures and analyses. I don't think they add value. You can mention that you get similar results with this different metric of variability. c. Figure S8-S10 can be replaced with a sentence or two after Figure 3 stating if the PDF changes are statistically significant or not (as mentioned in the comment about Figure 3). d. Figure S11 seems like a pretty critical figure. I don't think you can avoid this piece of the story! Figure S12 supports Figure S11, but is slightly less crucial.

Response: We accept this criticism. Following the suggestion of the reviewer, we substantially reduce the supplemental figures in the revised manuscript (please see the supplementary material). There are in total seven supplemental figures now.

3. Lines 105-6: This claim that DTDTx and DTDTn both increase in three seasons in the northern mid- to high-latitudes does not seem consistent with Figure 2. It seems correct for DTDTn, but DTDTx only shows broad increases in DJF. In MAM there is very little coherent signal north of 50N (and what there is blue). In SON, there is no high-latitude signal in North America or Asia, and only a small positive signal in eastern Europe. This picture is in conflict with the claim of "all models" agreeing (lines 109-112).

Response: In the revised manuscript, we show the variability of daily mean temperature

(DTDT) instead of the variabilities of daily maximum (DTDT_x) and minimum (DTDT_n) temperatures shown in the original manuscript (please see Fig. 1). We do this because the mechanisms are more appropriate to explain changes in DTDT. The DTDT signal is similar to the DTDT_x and DTDT_n signals. Accordingly, we have rewritten this paragraph:

Lines 138-146: Figs. 1c-1f show the biogeophysical effects of deforestation on DTDT for each season. The multimodel mean suggests a widespread increase in DTDT in deforested areas in the northern extratropics, e.g., North America and Eurasia. Such a DTDT signal is strongest (up to 0.7 °C or 20%) in winter, moderate in spring and autumn (between 0.2–0.4 °C or 10–15%), and weakest in summer (between 0.1–0.2 °C or 5–10%). More importantly, all five models agree well on the sign of the DTDT change (see the black dots), implying a robust signal across the models. In other deforested areas (e.g., tropics and the southern hemisphere), however, the DTDT response to deforestation is mostly negligibly small (within ±0.1 °C or ±5%) and has low intermodel consistency.

4. Figure 3 (and associated text): Have you verified the statistical significance of the differences between the PDFs shown in Figure 3? Can you quantify the percentage of the distribution that moves from the near-zero region to the tail region? My visual estimate is 3-4% of the distribution gets shifted from (-3,3) to the tails beyond this region. That may be statistically significant. If you can perform such tests on these pdfs, you can probably remove Figures S8-S10 and instead just mention if the PDF changes in each season are significant or not.

Response: We thank the reviewer for the constructive comment. Following the suggestion of the reviewer, we verify the statistical significance of the differences between the PDFs based on the Kolmogorov–Smirnov two-sample test in the revised manuscript. The result shows that the differences between the PDFs are mostly significant. Please see Fig. 2 and the revised manuscript:

Lines 196-200: Moreover, the difference in the probability density distribution of δT between the *piControl* and *deforest-globe* simulations is statistically significant at the 95% confidence level (see the p values in Fig. 2) tested by the Kolmogorov–Smirnov two-sample test (see “Methods”), except for Eurasia in summer.

Lines 738-743: **The Kolmogorov–Smirnov two-sample test.** The Kolmogorov–Smirnov two-sample test is a nonparametric test to determine if two samples of data are from the same continuous distribution. The null hypothesis is that the two dataset values are from the same continuous distribution. The alternative hypothesis is that these two datasets are from different continuous distributions. The hypothesis test is carried out at the confidence level of 95%. This test is performed using MATLAB.

Following the suggestion of the reviewer, we also quantify the percentage of the distribution that moves from the near-zero region to the tail region (please see Fig. 2) in the revised manuscript:

Lines 189-200: Compared to the *piControl* (blue bars), the frequency is lower for the near-zero region (e.g., $-3\text{ }^{\circ}\text{C} < \delta T < 3\text{ }^{\circ}\text{C}$) and higher for the tail regions (e.g., $\delta T < -3\text{ }^{\circ}\text{C}$ or $\delta T > 3\text{ }^{\circ}\text{C}$) for the *deforest-globe* (orange bars). Meanwhile, the frequency changes for the left and right tail regions are almost identical. Taking North America as an example, the frequency for wintertime δT within -3 to $3\text{ }^{\circ}\text{C}$ is decreased by 5.3%, whereas the frequency for wintertime δT below $-3\text{ }^{\circ}\text{C}$ and above $3\text{ }^{\circ}\text{C}$ is increased by 2.8% and 2.5%, respectively (Fig. 2a). This result implies an increased likelihood of both rapid cooling and warming events following deforestation.

In addition, we further analyze the percentage change in the frequency for the tail regions. We find that the percentage change nonmonotonically increases in the tail regions, implying larger influences of deforestation on the more extreme warming and cooling events. Please see Fig. 2, Fig. S6 and the revised manuscript:

Lines 201-212: Intuitively, the deforestation-induced increase in the frequency for the

tail regions seems small (no more than 6%). Given the low frequency for the tail regions, however, such an increase is not trivial. We further examine the percentage change ($\frac{\text{deforest-globe} - \text{piControl}}{\text{piControl}} \times 100\%$) in the probability density distribution of δT following deforestation (green lines in Fig. 2). The percentage change in the frequency for the tail regions is considerable and nonmonotonically increases with higher magnitudes of δT . For example, the frequencies of moderate ($-10\text{ }^\circ\text{C} < \delta T < -5\text{ }^\circ\text{C}$ or $5\text{ }^\circ\text{C} < \delta T < 10\text{ }^\circ\text{C}$) and extreme ($\delta T < -10\text{ }^\circ\text{C}$ or $\delta T > 10\text{ }^\circ\text{C}$) wintertime δT are increased by 12.8% and 34.1%, respectively, in North America. Note that such percentage changes in the frequency for the tail regions can be even larger at local scales (Fig. S6). This amplified increase in the frequency for the tail regions implies larger influences of deforestation on more extreme warming or cooling events.

5. Lines 198-200: You need to include descriptions of how you calculate each of these three terms. (Later I got to the methods and saw that the equations are provided in the methods: you need to mention this in the main text.) You need to convince the reader that these effects are easy to separate.

Response: *Nature Communications* requires to put the details of methods at the end of the main text. Thus, we can do nothing within this. In the revised manuscript, we have rewritten this part in the “Methods” section to give a more detailed description of the attribution method, including how the three terms are derived from the thermodynamic energy equation and the required model outputs to calculate the three terms. Please see the revised manuscript:

Lines 602-649: Attribution of the change in daily temperature variability. According to the thermodynamic energy equation⁵⁴, the temperature difference (δT) between two neighboring days can be expressed as:

$$\delta T = \frac{\partial T}{\partial t} = -\mathbf{u} \cdot \nabla T + \left(\frac{RT}{c_p P} - \frac{\partial T}{\partial P} \right) \omega + \frac{1}{c_p} \frac{\partial Q}{\partial t} \quad (3)$$

where T is the daily 2-meter temperature ($^\circ\text{C}$), u is the daily 10-meter vector wind ($\text{m}\cdot\text{s}^{-1}$)

1), ∇ is the horizontal divergence operator, R is the gas constant for dry air ($287 \text{ J}\cdot\text{K}^{-1}\cdot\text{kg}^{-1}$), c_p is the specific heat of dry air at constant pressure ($1004 \text{ J}\cdot\text{K}^{-1}\cdot\text{kg}^{-1}$), P is the atmospheric pressure (Pa), ω is the daily near-surface vertical velocity ($\text{Pa}\cdot\text{s}^{-1}$), and Q is the daily diabatic heating ($\text{J}\cdot\text{kg}^{-1}$).

Substituting Equation (3) into Equation (1), Equation (1) can be rewritten as:

$$\text{DTDT} = \frac{1}{n-1} \sum_{i=1}^{n-1} |\delta T_i| = \frac{1}{n-1} \sum_{i=1}^{n-1} \left| \frac{\partial T}{\partial t_i} \right| \approx \text{term1} + \text{term2} + \text{term3} \quad (4)$$

$$\text{term1} = \frac{1}{n-1} \sum_{i=1}^{n-1} |-\mathbf{u}_i \cdot \nabla T_i| \quad (5)$$

$$\text{term2} = \frac{1}{n-1} \sum_{i=1}^{n-1} \left| \left(\frac{RT_i}{c_p P_i} - \frac{\partial T}{\partial P} \right) \omega_i \right| \quad (6)$$

$$\text{term3} = \frac{1}{n-1} \sum_{i=1}^{n-1} \left| \frac{1}{c_p} \frac{\partial Q}{\partial t_i} \right| \quad (7)$$

Then, the deforestation-induced change in DTDT can be expressed as:

$$\Delta \text{DTDT} = \Delta \text{term1} + \Delta \text{term2} + \Delta \text{term3} \quad (8)$$

where Δ denotes the difference between the *piControl* and *deforest-globe* simulations (*deforest-globe* minus *piControl*) or between the *historical* and *hist-noLu* simulations (*historical* minus *hist-noLu*). Δterm1 denotes the change in the horizontal near-surface temperature advection. Δterm2 denotes the change in adiabatic compression and vertical advection. Δterm2 is commonly negligibly small in magnitude and can be reasonably omitted. Δterm3 denotes the change in the daily variability of diabatic heating.

For Δterm3 , diabatic heating is considered to be determined by surface sensible heat flux (SHF) because deforestation directly impacts near-surface temperature mainly through the change in SHF^{62, 63}. Strictly speaking, to calculate Δterm3 , the sensible heat absorbed by the 2-meter air should be used. However, only the total sensible heat that

warms the aboveground air is known. To compromise, we use the total sensible heat to qualitatively estimate the contribution of Δterm3 to the DTDT change, expressed as:

$$\Delta\text{term3} = \Delta\left(\frac{1}{n-1} \sum_{i=1}^{n-1} \left| \frac{\partial \text{SHF}}{\partial t} \right| \right) = \Delta\left(\frac{1}{n-1} \sum_{i=1}^{n-1} |\text{SHF}_{i+1} - \text{SHF}_i| \right) \quad (9)$$

where SHF_i denotes the surface sensible heat flux on day i . Obviously, Δterm3 is actually the change in *the day-to-day variation in SHF* (DTDSHF; see Equation (1)). Emphasize again that Δterm3 (Equation (9)) can only qualitatively explain the DTDT change, e.g., positive Δterm3 leading to increased DTDT.

In summary, the DTDT response to deforestation can be attributed to the change in the horizontal near-surface temperature advection (TADV) and the change in the daily variability of surface sensible heat flux (DTDSHF). Note that the effects of changes in albedo, surface roughness and evaporation efficiency that may arise from deforestation have already been covered by the TADV and DTDSHF effects. For example, the albedo-driven cooling effect and the evapotranspiration-driven warming effect determine the surface temperature change, which directly influences DTDSHF and indirectly influences the TADV through the modification of the horizontal temperature gradient. The lower surface roughness increases near-surface wind, favoring the higher TADV. Similarly, the atmospheric feedbacks^{64, 65} that arise from deforestation to surface temperature (e.g., cloud formation and downwards radiation) are also implicitly included in the TADV and DTDSHF effects.

The model output variables required to calculate TADV and DTDSHF are listed in Table S2.

6. Lines 204-6: You claim that Figure 4 shows that this mechanism largely explains the signal in Figure 2. But Figure 2 only shows a response in some seasons; Figure 4 shows nearly equal effects in all seasons. This seems to contradict your rationale. Your physical explanation associated with Figure 5 (through line 220) supports your claim, but the fact that this mechanism is equally active in the summer (when there

is no corresponding signal in Figure 2) makes it difficult to believe that you've captured the full story.

Response: We have updated the mechanisms in the revised manuscript. First, we find that the calculated horizontal temperature advection in the original manuscript is problematic due to a mistake in the code. Now, we have corrected the code. Second, in the original manuscript, we attributed the DTDT changes to the horizontal temperature advection at the 850 hPa layer. We find this is improper because, following the attribution method, the DTDT change should be explained by the horizontal temperature advection at the same layer. We, therefore, attribute the DTDT change to near-surface, instead of 850 hPa, horizontal temperature advection in the revised manuscript. The updated temperature advection effect and the sensible heat effect can reasonably explain the seasonal variation in daily temperature variability. In the revised manuscript, we also additionally provide observational evidence from 24 paired sites of spatially adjacent forest and openland sites to support the mechanisms identified in the simulations.

Please see Fig. 3, Fig. 4, Fig. S5 and the revised manuscript:

Lines 226-298: Mechanisms for the deforestation effects on daily temperature variability. Explaining the change in temperature variability is always challenging, and we attempt to reveal the mechanisms for the deforestation effects on DTDT based on the thermodynamic energy equation⁵⁴. An increase in DTDT can be attributed to 1) an increase in the day-to-day variation in surface sensible heat flux (DTDSHF) and/or 2) an increase in near-surface horizontal temperature advection (TADV; see Methods).

Fig. 3 shows the DTDSHF and TADV responses to deforestation for each season. Unfortunately, only one model (CanESM5) provides the required daily 10-meter wind speed to calculate TADV; thus, the TADV response is from the CanESM5 model. As shown in Fig. 3, DTDSHF and TADV show contrasting responses to deforestation. The models consistently suggest a decrease in DTDSHF in almost all deforested areas (Figs.

3a-3d). The DTDSHF change is larger (up to $-7 \text{ W}\cdot\text{m}^{-2}$) in spring and summer but small (mostly within $-3 \text{ W}\cdot\text{m}^{-2}$) in winter and autumn in the northern extratropical deforested areas. For the tropical deforested areas, however, the DTDSHF change is moderate (-5 to $-2 \text{ W}\cdot\text{m}^{-2}$) and stable throughout the year. Such a decrease in DTDSHF can be explained by the decreased surface aerodynamic roughness in deforested areas. The lower roughness causes the surface to be less efficient in warming the aboveground atmosphere through turbulence, suppressing the daily variability in sensible heat flux. The daily variability of sensible heat flux can be alternatively quantified by the standard deviation index (SDSHF), and a similar result can be obtained (not shown). The DTDSHF response to deforestation is also supported by the observational evidence from the paired sites in North America and Europe (Figs. S5d).

In contrast to the DTDSHF change, the CanESM5 model suggests a widespread increase in TADV in most deforested areas and neighboring regions (Figs. 3e-3 h). The TADV signal is stronger ($> 0.4 \text{ }^\circ\text{C}$) in the northern extratropical deforested areas, with the largest magnitude in winter, followed by spring and autumn, and the lowest magnitude in summer. The TADV signal is mostly weaker (within $0.2 \text{ }^\circ\text{C}$) in the deforested areas in the tropics and the southern hemisphere, except for some marginal areas of the Amazon basin and the Indo-China Peninsula, throughout the year.

The DTDSHF and TADV changes can basically explain the DTDT response to deforestation. For the northern extratropical deforested areas, the enhanced TADV, despite being slightly offset by the decreased DTDSHF, dominates the DTDT increase. Meanwhile, the TADV signal is stronger in winter, spring and summer, whereas the DTDSHF signal is stronger in spring and summer, causing seasonal variations in DTDT responses to deforestation there. For example, the larger increase in wintertime DTDT is a consequence of the larger increase in wintertime TADV and the small change in wintertime DTDSHF. For the tropical deforested areas, however, the TADV change is small and mostly offset by the decreased DTDSHF. Moreover, the climatological mean DTDT is smaller in the tropics than in boreal regions. As a result, the DTDT change is

negligibly small in the tropical deforested areas.

As mentioned earlier, the observed DTDT signal from the paired sites (Fig. S5c) is inconsistent with the simulated signal (Figs. 1c-1f) in spring, summer and autumn. This discrepancy can be explained by the TADV response. Specifically, the atmosphere above the paired forest and openland sites tends to be mixed due to horizontal advection, alleviating the TADV difference between the paired sites. Meanwhile, the DTDSHF response to deforestation is larger in spring, summer and autumn (Figs. 3a-3d and Fig. S5d). Thus, the DTDT differences between the paired sites are more influenced by the DTDSHF effect except in winter. That is why the observations suggest an overall lower DTDT over the openland sites in these seasons (Fig. S5c), contrasting with the simulated DTDT change.

While the result on the TADV response to deforestation is from a single model, this result is also indirectly supported by other models and the observations from the paired sites. An increase in TADV can be caused by an increase in near-surface wind speed or horizontal temperature gradient (see Methods). Fig. 4 shows the responses of the near-surface wind speed and horizontal temperature gradient to deforestation for each season. All the models (Figs. 4a-4d) and the in situ observations (Figs. S5e) consistently suggest an increase in near-surface horizontal wind speed due to the lower surface roughness in deforested areas or the openland sites, which tends to increase TADV. The models also consistently suggest an enhanced near-surface horizontal temperature gradient in some midlatitude deforested areas in winter and spring (Figs. 4e-4h), also favoring the increase in TADV. This enhanced horizontal temperature gradient can be explained by the strong cooling effect of deforestation in boreal regions (e.g., Fig. S3).

Furthermore, another review suggests that we investigate the effects of historical deforestation in addition to idealized deforestation. We also find that the mechanisms for the effects of historical deforestation are highly consistent with those for idealized deforestation. Please see the revised manuscript:

Lines 327-341: As a result of such large-scale deforestation, most models simulate increases in wintertime DTD (with multimodel mean values of 0.2 - 0.5 °C or 2 - 12%; Fig. 5c) and SDT (with multimodel mean values of 0.2 - 0.6 °C or 2 - 10%; not shown). In other words, historical deforestation has led to a detectable increase in the wintertime daily temperature variability in North America. The enhanced daily temperature variability can be attributed to the higher TADV (Fig. 5e) and simultaneously is slightly offset by the lower DTDSHF (Fig. 5d). These mechanisms are consistent with those for the deforestation effects shown in the idealized deforestation scenario. Unfortunately, again only one model (ACCESS-ESM1-5) supports the analysis of the TADV response to the historical deforestation here due to the data availability. Nevertheless, the wintertime TADV response to deforestation shown by the ACCESS-ESM1-5 model (Fig. 5e) is broadly consistent with the CanESM5 model (Fig. 3e). More importantly, the consistent DTD or SDT responses to the idealized deforestation scenario and the historical deforestation scenario as well as the consistent mechanisms consolidate the robustness of the results.

7. Line 286: I'm troubled by "we do not question the rationality of the simulated cooling effects of deforestation." Shouldn't you always question the rationality of the mean behavior of the primary variable you are investigating? As I said with regard to Figure S11, I don't think you can avoid this part of the story!

Response: In the revised manuscript, we evaluate the model representation of the deforestation effects on mean temperature with satellite observations following the suggestion. The results show that the models are basically reasonable in this regard. Please see Fig. S2 and the revised manuscript:

Lines 165-178: We also evaluate the model representation of the biogeophysical effects of deforestation on mean surface temperature with satellite observations^{13, 48} (see "Methods"). The models can basically reproduce the latitude-dependent responses of mean surface temperature to deforestation shown by the satellite observations (Fig. S2) and the previous studies⁶⁻⁹. Note that, in contrast to the satellite observations, the

models simulate a more widespread cooling effect of deforestation in the northern mid-latitudes. This discrepancy is largely attributed to the fact that the deforestation-induced atmospheric feedbacks to surface temperature are fully considered in the models but mostly absent in the observed signal^{49,50} (see “Methods”).

Lines 679-712: Validation of the model representation of the biogeophysical effects of deforestation on mean surface temperature. To evaluate the model behavior in the representation of the biogeophysical effects of afforestation on mean surface temperature, a dataset⁴⁸ mapping the biophysical effects of vegetation cover change at the global scale is used. This dataset is established based on the “space-for-time” substitution method and multiple satellite observations. We use the dataset mapping the potential changes in daytime (~ 13:30 local time) and nighttime (~ 01:30 local time) surface temperatures as a result of a complete transition from forests to grasslands and croplands. The dataset is provided at monthly time steps and a horizontal resolution of $1^\circ \times 1^\circ$. The monthly values are averaged over the period of 2008-2012. The daytime and nighttime values are averaged to obtain the estimate of the daily mean value.

Compared to the satellite observations, the models simulate more widespread cooling effects in the northern mid-latitudes (Fig. S2). It should be emphasized that such discrepancies cannot be fully attributed to the poor performance of the models. Instead, this discrepancy largely results from the unfair comparison between the observed and simulated biogeophysical effects⁵⁰. Specifically, the observed biogeophysical effects of deforestation are retrieved based on the “space-for-time” substitution method through the comparison of the spatially adjacent forest and openland pixels. Since the atmosphere is assumed to be largely identical over the forest pixels and the neighboring openland pixels, the deforestation-induced atmospheric feedbacks to surface temperature (e.g., cloud formation and downwards radiation)^{64, 65} are not fully considered. Thus, the observed biogeophysical effects should be considered largely representative of the local effects of deforestation^{13, 48}. In contrast, the atmospheric feedbacks to deforestation are fully considered in the coupled models. Thus, the

simulated biogeophysical effects include both the local and nonlocal (through atmospheric feedbacks) effects of deforestation⁴⁹. It has been explicitly demonstrated that deforestation causes widespread nonlocal cooling effects in the northern mid- to high latitudes, with magnitudes comparable to or even larger than the local effects^{49, 66}. That is why the simulations indicate a more widespread cooling effect of deforestation in the northern mid-latitudes than the satellite observations^{49, 50} (Fig. S2). Moreover, the biogeophysical effects of deforestation also depend on the background climate^{67, 68}. Thus, the cold bias in the simulated biogeophysical effects can also be partly attributed to the colder background climate of the *piControl*, with the anthropogenic forcings fixed at the levels of year 1850.

8. I expected Figure S13 to be in equivalent units as Figure 4 since you describe them both as “the effect of ...”. In the text you seem to downplay the sensible heat flux effect because it is strongest in JJA and SON. You should present these two figures in comparable ways so we can see if perhaps the summer-time effect in Figure S13 counters the thermal advection affect shown in all seasons in Figure 4. That might be the answer to the fundamental unanswered question I have about this paper!

Response: We thank the reviewer for the constructive comment. We have moved the figure showing the sensible heat flux effect from the supplementary material to the main text. As the reviewer said, the sensible heat flux effect plays an important role in regulating the seasonal variation in the response of daily temperature variability to deforestation. Please see Fig. 3 and the revised manuscript:

Lines 235-247: The models consistently suggest a decrease in DTDSHF in almost all deforested areas (Figs. 3a-3d). The DTDSHF change is larger (up to $-7 \text{ W}\cdot\text{m}^{-2}$) in spring and summer but small (mostly within $-3 \text{ W}\cdot\text{m}^{-2}$) in winter and autumn in the northern extratropical deforested areas. For the tropical deforested areas, however, the DTDSHF change is moderate (-5 to $-2 \text{ W}\cdot\text{m}^{-2}$) and stable throughout the year. Such a decrease in DTDSHF can be explained by the decreased surface aerodynamic roughness in deforested areas. The lower roughness causes the surface to be less efficient in warming

the aboveground atmosphere through turbulence, suppressing the daily variability in sensible heat flux. The daily variability of sensible heat flux can be alternatively quantified by the standard deviation index (SDSHF), and a similar result can be obtained (not shown). The DTDSHF response to deforestation is also supported by the observational evidence from the paired sites in North America and Europe (Figs. S5d).

Lines 266-276: The DTDSHF and TADV changes can basically explain the DTDT response to deforestation. For the northern extratropical deforested areas, the enhanced TADV, despite being slightly offset by the decreased DTDSHF, dominates the DTDT increase. Meanwhile, the TADV signal is stronger in winter, spring and summer, whereas the DTDSHF signal is stronger in spring and summer, causing seasonal variations in DTDT responses to deforestation there. For example, the larger increase in wintertime DTDT is a consequence of the larger increase in wintertime TADV and the small change in wintertime DTDSHF. For the tropical deforested areas, however, the TADV change is small and mostly offset by the decreased DTDSHF. Moreover, the climatological mean DTDT is smaller in the tropics than in boreal regions. As a result, the DTDT change is negligibly small in the tropical deforested areas.

The reviewer also suggested that we show the temperature advection effect and the sensible heat flux effect in equivalent units. Unfortunately, this is difficult to do because we only know the total sensible heat flux but do not know the sensible heat flux absorbed by the 2-meter air required by Equation (3). Please see the revised manuscript:

Lines 625-635: For Δterm_3 , diabatic heating is considered to be determined by surface sensible heat flux (SHF) because deforestation directly impacts near-surface temperature mainly through the change in SHF^{62, 63}. Strictly speaking, to calculate Δterm_3 , the sensible heat absorbed by the 2-meter air should be used. However, only the total sensible heat that warms the aboveground air is known. To compromise, we use the total sensible heat to *qualitatively* estimate the contribution of Δterm_3 to the DTDT change, expressed as:

$$\Delta_{\text{term3}} = \Delta\left(\frac{1}{n-1} \sum_{i=1}^{n-1} \left| \frac{\partial SHF}{\partial t} \right| \right) = \Delta\left(\frac{1}{n-1} \sum_{i=1}^{n-1} |SHF_{i+1} - SHF_i| \right) \quad (9)$$

where SHF_i denotes the surface sensible heat flux on day i . Obviously, Δ_{term3} is actually the change in *the day-to-day variation in SHF* (DTDSHF; see Equation (1)). Emphasize again that Δ_{term3} (Equation (9)) can only qualitatively explain the DTDT change, e.g., positive Δ_{term3} leading to increased DTDT.

We acknowledge this uncertainty in the mechanisms and provide a discussion on this point in the revised manuscript:

Lines 444-454: We acknowledge the uncertainty in the mechanisms for the biogeophysical effects of deforestation on daily temperature due to the model output availability. In particular, the effect related to horizontal near-surface temperature advection is examined based on limited models. It should be emphasized that, despite this limitation, the deforestation effects on daily temperature variability are robust because the models and observations consistently agree on these effects, and the results are independent of the metrics of the daily temperature variability. Since the daily variables are required to study the daily temperature variability, we encourage more model groups that contribute to the CMIP6 and LUMIP to report daily output to support further analysis. We also encourage other groups to study the mechanisms for the deforestation effects on daily temperature variability with their own models and experiments.

Reviewer #3 (Remarks to the Author):

In their study, Ge et al. analyze the effect of deforestation on the day-to-day (DTD) variations in surface air temperature for different seasons using Earth system model (ESM) simulations from the Land Use Model Intercomparison Project LUMIP, part of CMIP6. The authors find that the DTD variability increases for daily maximum and minimum temperature, but the changes are only noteworthy during the winter months in North America and Eurasia. By decomposing a thermodynamic equation using the model output of one ESM, they disentangle the different effects contributing to the increased variability. They find that increased horizontal advection of warm air is the main cause for increased DTD variability rooted in increased meridional pressure and temperature gradients.

While the study and its results are interesting, the drawn conclusions are overstated. Please see my General Comments for details.

Response: We thank the reviewer for the positive comments. We also accept the criticism that we may overstate the conclusion in the original manuscript. We delete the sentences that overstate the conclusion in the revised manuscript. Please see the detailed response to the general comment 2 below.

Also, the model simulations used in this study are quite idealized, mainly because all other historical drivers that impact on DTD variability are switched off in the model setup.

Response: We agree that the deforestation scenario used in the original manuscript is quite idealized. In the revised manuscript, we further analyze the effects of historical deforestation on daily temperature variability in addition to the idealized deforestation scenario. We find that the deforestation effects are comparable to the effects of other anthropogenic forcings (e.g., GHGs and aerosols) at local and regional scales. Please see the detailed response to the general comment 1 below.

The found effect might vanish when the other drivers are considered. This aspect needs to be addressed, because the narrative of the manuscript stresses the importance of this effect and “its broad impacts on public health, ecosystem functions and the economy”. Overall, the structure of the manuscript is clear and concise, the English language, however, needs some proofreading, especially towards logic linking of sentences.

Bottom line, the study is definitely interesting, but the findings are not strong and significant enough to back the conclusions drawn. In its current state, I cannot recommend the manuscript for publication in Nature Communications. I, however, do recommend and encourage publication in a more specialized community journal after some revisions. Please see my comments below.

1 General Comments:

1.1 I wonder whether the signals you find are really noteworthy. First, the strength of the deforestation effect on the DTD variability should be compared to the effect strength of other drivers, such as rising CO₂. This would help to get an idea whether this effect is significant enough and actually provides enough support to your conclusions, e.g., the statement, that the effects of forest changes on daily temperature variability need to be considered when implementing large-scale afforestation to mitigate climate change. Further, in your Figures S1 and S2 you present a comparison of your variability metrics between observations/reanalysis and the multi-model mean. Based on this results you argue “Validated against the observational and the reanalysis data, the five models can well reproduce the spatial and temporal variations....” And further “The reasonable simulations ... guarantee the rationality of further studies based on the models.” I am not convinced that the models reproduce the reanalysis and observations well, especially in the regions and seasons where you find the strongest signal, so cold season and high-mid latitudes. In comparison to HadGHCND there is a bias of at least 3° C in the models for DJF and North America/Eurasia. Comparing to reanalysis, the bias is very strong in SON, where models underestimate DTD variability by a few degrees. The

largest change due to deforestation you find is around 0.5 °C. Given the high uncertainty, the signal due to deforestation might be over-interpreted and conclusion far-fetched.

Response: There are two comments here. First, the reviewer suggests that we compare the effect of deforestation with the effects of other anthropogenic forcings. To this end, we investigate the biogeophysical effects of deforestation during the historical period (1850-2014) with the *historical* simulation from the CMIP6 and the *hist-noLu* simulation from the LUMIP (*historical* minus *hist-noLu*). We also investigate the effects of all anthropogenic forcings during the historical period (1850-2014) with the *historical* simulation from the CMIP6 and the *hist-nat* simulation from the DAMIP (*historical* minus *hist-noLu*). We find that the deforestation effects are comparable to the effects of other anthropogenic forcings (e.g., GHGs and aerosols) at local and regional scales.

Please see Fig. 5 and the revised manuscript:

Lines 307-364: **The effects of historical deforestation on daily temperature variability.** The above analysis is built on the idealized deforestation scenario, which is unlikely to occur in the real world. We therefore wonder whether the real-world deforestation over the historical period would have generated a detectable signal as well. To this end, the historical simulation (*historical*) of the CMIP6 and the historical no land-use simulation (*hist-noLu*) of the LUMIP are used. The *historical* covers the period of 1850–2014 with evolving and externally imposed natural and anthropogenic forcings (e.g., GHGs, aerosols and LULCCs; see “Methods”). The *hist-noLu* is identical to the *historical*, except that LULCCs are fixed at preindustrial levels (see “Methods”). The last 30-year simulations (1985–2014) of the *historical* and *hist-noLu* are compared (*hist-noLu* minus *historical*) to isolate the deforestation effect. Nine earth system models are used here given the model output availability (Table S1). These models are also reasonable in the DTD simulations validated against the reanalysis (Figs. S1e-1h and S1i-1p; see “Methods”).

Tree cover changes during the historical period are highly heterogeneous in space with the concurrence of forest losses and gains (Fig. S7), which may confound the signals arising from forest losses and gains. Exceptionally, we find tremendous and spatially coherent tree cover losses in North America since 1850 (Fig. 5a). Moreover, this decreasing trend of tree cover is also consistent across the models, with the multimodel mean value of the net tree cover loss being approximately -1.3 million km² during the historical period of 1850-2014 (Fig. 5b). As a result of such large-scale deforestation, most models simulate increases in wintertime DTDT (with multimodel mean values of 0.2 - 0.5 °C or 2 - 12%; Fig. 5c) and SDT (with multimodel mean values of 0.2 - 0.6 °C or 2 - 10%; not shown). In other words, historical deforestation has led to a detectable increase in the wintertime daily temperature variability in North America. The enhanced daily temperature variability can be attributed to the higher TADV (Fig. 5e) and simultaneously is slightly offset by the lower DTDSHF (Fig. 5d). These mechanisms are consistent with those for the deforestation effects shown in the idealized deforestation scenario. Unfortunately, again only one model (ACCESS-ESM1-5) supports the analysis of the TADV response to the historical deforestation here due to the data availability. Nevertheless, the wintertime TADV response to deforestation shown by the ACCESS-ESM1-5 model (Fig. 5e) is broadly consistent with the CanESM5 model (Fig. 3e). More importantly, the consistent DTDT or SDT responses to the idealized deforestation scenario and the historical deforestation scenario as well as the consistent mechanisms consolidate the robustness of the results.

Note that there is some spatial mismatch between the DTDT change (Fig. 5c) and the TADV change (Fig. 5e). This mismatch is because the pattern of tree cover changes for the ACCESS-ESM1-5 model (Fig. S7a) is slightly different from the multimodel mean pattern (Fig. 5a).

We further examine whether the deforestation effects on daily temperature variability are noteworthy compared to the effects of other anthropogenic forcings (e.g., GHGs and aerosols). To this end, the historical natural-only simulation (*hist-nat*) of the

Detection and Attribution Model Intercomparison Project⁵⁵ (DAMIP) is used. The *hist-nat* is identical to the *historical* but is imposed by only natural (e.g., solar and volcanic aerosols) forcings (see “Methods”). Accordingly, comparing the last 30-year (1985–2014) simulations of the *historical* and *hist-nat* (*historical* minus *hist-nat*) isolates the net effect of all anthropogenic forcings (GHGs, aerosols and LULCCs) during the historical period.

At the global scale, the deforestation effects on daily temperature variability are much smaller than the effects of other anthropogenic forcings (not shown). At local and regional scales, however, the deforestation effects can be comparable to the other anthropogenic forcings in magnitude, particularly in deforested areas. In North America, for example, the wintertime DTDT shows an overall decreasing trend during the historical period with all anthropogenic forcings considered (Fig. 5f). This decreasing trend in daily temperature variability is also supported by the previous studies³⁴⁻³⁷. Interestingly, this decreasing trend in DTDT is mostly absent in the deforested area (Fig. 5a and 5f), implying that the positive effect of deforestation (Fig. 5c) largely offset the effects of other anthropogenic forcings (GHGs and aerosols).

Lines 500-546: **Investigation of the biogeophysical effect of historical deforestation.** To obtain the biogeophysical effects of deforestation during the historical period, the historical simulation (*historical*) of the CMIP6⁴³ and the historical no land-use simulation (*hist-noLu*) of the LUMIP⁴⁴ are used.

The *historical* is a fully coupled simulation covering the historical period of 1850-2014. This simulation is branched from the *piControl* and is driven by evolving and externally imposed forcings. All natural (e.g., volcanic aerosols and solar variability) and anthropogenic (e.g., GHGs, aerosols and LULCCs) forcings are included and are largely based on observations. The historical LULCCs are prescribed by the new generation of the Land-Use Harmonization 2 (LUH2) dataset⁶¹ and include detailed changes in land cover, land use and land management.

The *hist-noLu* is the same as the *historical* except with LULCCs fixed at the levels of year 1850. Note that the atmospheric CO₂ concentration in the *hist-noLu* is identical to the value in the *historical*. Thus, the biogeophysical effect of historical deforestation is isolated by comparing the last 30-year (1985-2014) simulations of the *historical* and *hist-noLu* (*historical* minus *hist-noLu*). By the time of this study, nine earth system models contribute to both the *historical* and *hist-noLu* simulations and provide the required daily 2-meter temperature output; thus, these nine models are used here (Table S1). For each model, multiple members are available for the *historical* and *hist-noLu*. To save computational cost, the first member of the *historical* and *hist-noLu* are used for analysis.

The total tree cover change during the historical period is obtained by subtracting the last 30-year (1985-2014) mean tree cover fraction of the *hist-noLu* from the corresponding value of the *deforest-globe*. Note that while the historical LULCCs are prescribed by the LUH2, the implementation of the historic LULCCs across different models is commonly different (Fig. S7), and this problem has always plagued model intercomparison⁴⁴. Such intermodel consistency may cause the models to diverge in the response of daily temperature variability to the historical LULCCs. Moreover, both tree losses and gains may occur in a region, confounding the signals in daily temperature variability arising from forest losses and gains. Fortunately, we find a large area of forest losses during the historical period in North America (Fig. 5a). This tree cover change is considerable, spatially coherent and of high intermodel consistency. Therefore, we take North America as a case study to investigate the response of daily temperature variability to historical deforestation.

Investigation of the net effect of anthropogenic forcings. To obtain the effects of anthropogenic forcings during the historical period of 1850-2014, the *hist-nat* simulation of the Detection and Attribution Model Intercomparison Project⁵⁵ (DAMIP), as well as the *historical*, are used.

The *hist-nat* resembles the *historical* but is forced with only solar and volcanic forcings

from the *historical* simulation. Thus, the net effect of all anthropogenic forcings (GHGs, aerosols and LULCCs) is isolated by comparing the last 30-year (1985-2014) simulations of the *historical* and *hist-nat* (*historical* minus *hist-nat*). By the time of this study, four earth system models contribute to both the *historical* and *hist-nat* simulations and provide the required daily 2-meter temperature output; thus, these four models are used here (Table S1). For each model, multiple members are available for the *historical* and *hist-nat*. Similarly, the first member of the *historical* and *hist-nat* are used for analysis.

Second, the reviewer thinks that the models are not reasonable in the simulation of daily temperature variability. In the original manuscript, we use the HadGHCND dataset and NCEP-NCAR reanalysis as the observational benchmarks. However, we find that the use of these two datasets might be problematic. The HadGHCND dataset was produced based on station observations and spatial interpolation. We find considerable missing data in some grids for the HadGHCND dataset, probably leading to a larger uncertainty in this dataset. The NCEP-NCAR reanalysis was produced in the 1990s and is out of date. In the revised manuscript, we replace the HadGHCND dataset and NCEP-NCAR reanalysis with the ERA5 and NCEP-DOE AMIP-II reanalysis datasets. The ERA5 is the state-of-the-art reanalysis product from ECMWF. The NCEP-DOE AMIP-II is the new generation of the NCEP-NCAR reanalysis. The ERA5 and NCEP-DOE AMIP-II reanalysis datasets have also been used to study daily temperature variability in recent studies (e.g., Blackpot et al. 2021).

Please see Fig. S1, S3 and the revised manuscript:

Lines 121-128: The model performance in the DTDT simulation is first evaluated with two dependent reanalysis datasets: the ERA5 reanalysis⁴⁶ and the NCEP-DOE AMIP-II reanalysis⁴⁷ (see “Methods”). Validated with the reanalysis, the models can reasonably simulate the spatial pattern of DTDT with larger values at higher-latitude continents (Figs. S1a-1d and S1i-1p). Moreover, DTDT is overall larger in winter (December, January and February), followed by spring (March, April and May) and

autumn (September, October and November), and lowest in summer (June, July and August) in the northern extratropics.

Lines 160-161: The model behavior in the SDT simulation is also reasonable when validated against the reanalysis (Figs. S3a-3d and S3i-3p).

Lines 650-678: Validation of the model simulation of daily temperature variability.

To validate the model behavior in the simulations of DTDT and SDT, two dependent reanalysis datasets are used: the ERA5 reanalysis⁴⁶ from the European Center for Medium Weather Forecasting and the NCEP-DOE AMIP-II reanalysis⁴⁷ from the National Centers for Environmental Prediction (NCEP) and National Center for Atmospheric Research (NCAR). These two reanalysis datasets can provide reliable observations of daily temperature variability³⁵ and are widely used in the previous studies^{35, 37}. The ERA5 reanalysis provides the global daily mean 2-meter temperature at a horizontal resolution of 0.25° (latitude) \times 0.25° (longitude) since 1950. The NCEP-DOE AMI-II reanalysis provides global daily mean 2-meter temperature on the T62 Gaussian grid (192 longitude grids \times 94 latitude grids) since 1979.

The 30-year (1985–2014) mean DTDT and SDT from the ERA5 reanalysis (Figs. S1i-1l and S3i-3l) and the NCEP-DOE AMIP-II reanalysis (Figs. S1m-1p and S3m-3p) are used as the observational benchmarks. The 30-year (year 51–80) mean DTDT and SDT simulated by the *piControl* (Figs. S1a-1d and S3a-3d) and the 30-year mean (1985–2014) DTDT and SDT simulated by the *historical* (Figs. S1e-1h and S3e-3h) are validated against the reanalysis. Note that the anthropogenic forcings for the *piControl* are fixed at the levels of year 1850, which are inconsistent with the reanalysis levels. Thus, there might be larger biases in DTDT and SDT simulated by the *piControl* than the *historical*.

Validated with the reanalysis, both the *piControl* and *historical* can reasonably simulate the global patterns of DTDT and SDT for each season. Statistically speaking, compared to the ERA5 reanalysis, the root mean squared error (RMSE) in the simulated DTDT is

between 0.17–0.35 °C for the *piControl* and between 0.16–0.24 °C for the *historical* across the seasons. When compared to the NCEP-DOE AMIP-II reanalysis, the RMSE in DTD is slightly larger for both the *piControl* (0.25–0.37 °C across the seasons) and *historical* (0.23–0.47 °C across the seasons). In general, the RMSE in the simulated DTD is highest in winter, followed by spring and autumn, and lowest in summer. A similar result can be obtained for the simulated SDT.

1.2 I do not see that the increase in day-to-day temperature variability due to deforestation in the cold season in the mid- to high-latitudes has a “broad effect on public health, ecosystem functions and the economy” (L24). However, this is the central conclusion of your study. The strongest increase in DTD variability is around 0.5 °C in winter months according to your Figure 2. The ecosystems functions are likely not affected by these small changes, especially because most of the ecosystems there are in their dormant phase in winter. Also economic growth is likely not being affected by this small increase in DTD variability. You cite Kotz et al. (2021) here, who actually write that there is no effect of winter DTD variability on economic growth rates. Regarding public health you state “the reduced daily temperature variability in cold seasons is commonly beneficial to public health as people are in particular susceptible to a variety of disease in these seasons” and you cite the reference which is listed under number 39. Reference 39 is the LUMIP model experiment design paper. I doubt that an increase of 0.5 °C in DTD variability in winter has a measurable effect on human health – in summer maybe, but not in winter.

Response: We accept this criticism that we may overstate the impacts of changes in daily variability that arise from deforestation. In the revised manuscript, we have removed the “broad effect on public health, ecosystem functions and the economy” from the conclusions in the abstract and the main text. For example:

Lines 31-33: This study provides new insight into the deforestation or afforestation effects on temperature variability and has implications for the northern extratropical

countries that plan to implement large-scale afforestation.

Lines 94-98: This study fills the knowledge gap regarding the deforestation effects on temperature variability. On the other hand, this study provides new insight into human influences on daily temperature variability through deforestation or afforestation, which has implications for policy decisions on implementing large-scale afforestation.

While the DTDT signal is strongest in winter, DTDT changes are not negligibly small in other seasons. These DTDT changes may have potential impacts on human and natural systems. We acknowledge that we do not quantify the impacts of the DTDT changes due to deforestation on ecosystem function, economic growth or public health. This is definitely an interesting question that merits further investigation but goes out of the scope of this study. Instead, we mainly report the impacts of deforestation on daily temperature, which were rarely focused on before and may have broad impacts on human and natural systems.

We also delete the sentence “the reduced daily temperature variability in cold seasons is commonly beneficial to public health as people are in particular susceptible to a variety of disease in these seasons” in the revised manuscript.

1.3 I find the decomposition using thermodynamic equation an interesting and important analysis. Can you elaborate a bit on where the effect of surface roughness changes and changes in evaporative cooling enter the equation? Are these processes already covered by the first and third term, so by looking at horizontal advection and the estimates of surface sensible heat fluxes? Is it really enough to look into surface sensible heat fluxes when studying the diabatic heating term? Deforestation is associated with concomitant changes in surface albedo and surface latent heat fluxes - should that not be accounted for in your analysis? Some studies also show that deforestation can influence low-level cloud cover impacting on the amount of incoming short-wave and outgoing long-wave radiation. This again can modulate day-to-day variability of the surface air temperature. Could this cloud feedback play

a role here, too, as proposed by Duveiller et al. (2021)? The decomposition is only done for one model (CanESM5). The dynamics between the different GCMs + land-surface models are quite pronounced, so are the results reliable if they are only based on one model? And if so, why would CanESM5 be a good model to study this decomposition besides the fact that it is the only model for which the necessary output is accessible.

Response: We thank the reviewer for the constructive comments. As the reviewer said, the effects of changes in albedo, surface roughness and evaporation efficiency that may arise from deforestation have already been covered by the TADV and DTDSHF effects. Following the suggestion of the reviewer, we elaborate on this point in the revised manuscript:

Lines 636-647: In summary, the DTDT response to deforestation can be attributed to the change in the horizontal near-surface temperature advection (TADV) and the change in the daily variability of surface sensible heat flux (DTDSHF). Note that the effects of changes in albedo, surface roughness and evaporation efficiency that may arise from deforestation have already been masked by the TADV and DTDSHF effects. For example, the albedo-driven cooling effect and the evapotranspiration-driven warming effect determine the surface temperature change, which directly influences DTDSHF and indirectly influences the TADV through the modification of the horizontal temperature gradient. The lower surface roughness increases near-surface wind, favoring the higher TADV. Similarly, the atmospheric feedbacks^{64, 65} that arise from deforestation to surface temperature (e.g., cloud formation and downwards radiation) are also implicitly included in the TADV and DTDSHF effects.

We also cite the Duveiller et al. (2021) recommended by the reviewer in the revised manuscript:

Lines 898-899: Duveiller, G. et al. Revealing the widespread potential of forests to increase low level cloud cover. *Nat. Commun.* **12**, 4337 (2021).

2 Specific Comments:

2.1 L14: What about surface roughness changes due to deforestation?

Response: We have written this sentence in the revised manuscript given the word limit:

Lines 21-22: While biogeophysical effects of deforestation on temperature are broadly documented, how deforestation influences temperature variability remains largely unknown.

2.2 L17: “Show” is a strange word here. I would rather write: “Here, we investigate / analyze ...”

Response: Following the suggestion of the reviewer, we change “show” to “investigate” in the revised manuscript (please see line 23).

2.3 L18: There is only ‘the global scale’, not multiple global scales.

Response: Following the suggestion of the reviewer, we change “global scales” to “the global scale” in the revised manuscript (please see line 23).

2.4 LL30-31: Leaf area index and surface albedo is related, however, evapotranspiration should be mentioned here.

Response: Following the suggestion of the reviewer, we have written this sentence in the revised manuscript:

Lines 36-39: In addition to the impact on the carbon cycle, deforestation has profound impacts on local and regional climate through biogeophysical processes, e.g., higher surface albedo and lower aerodynamic roughness and evapotranspiration efficiency³⁻⁵.

2.5 LL43-45: That does not make logical sense. Why “but some studies, in contrast...”?

Where is the contrast?

Response: We have written this sentence:

Lines 49-50: This effect, however, is uncertain because some studies also report alleviated hot extremes following deforestation^{15, 16, 19}.

2.6 L47: What is a 'temperature series'? You mean 'time series' of temperature, right?

Response: Following the suggestion of the reviewer, we change "temperature series" to "a time series of temperature" in the revised manuscript (please see line 54).

2.7 LL84-85: In the literature I mostly see Tmax and Tmin as abbreviations. And what about daily mean temperatures?

Response: In the revised manuscript, we show the variability of daily mean temperature (DTDT) instead of the variabilities of daily maximum (DTDTx) and minimum (DTDTn) temperatures shown in the original manuscript.

2.8 L118: I would not say that these are different results. You just present the same results in a different way (absolute versus relative changes).

Response: In the revised manuscript, we only show the absolute changes, and we have rewritten this sentence:

Lines 140-142: Such a DTDT signal is strongest (up to 0.7 °C or 20%) in winter, moderate in spring and autumn (between 0.2–0.4 °C or 10–15%), and weakest in summer (between 0.1–0.2 °C or 5–10%).

2.9 Non-local effects are captured by the observations - we just cannot disentangle them from local effects or other drivers such as the greenhouse gas effect, etc.

Response: We agree with this comment, and we reorganize these sentences in the revised manuscript, for example:

Lines 135-137: This discrepancy is largely attributed to the fact that the deforestation-induced atmospheric feedbacks to surface temperature are fully considered in the models but mostly absent in the observed signal^{49, 50} (see “Methods”).

Lines 696-700: Since the atmosphere is assumed to be largely identical over the forest pixels and the neighboring openland pixels, the deforestation-induced atmospheric feedbacks to surface temperature (e.g., cloud formation and downwards radiation)^{64, 65} are not fully considered.

2.10 LL399: Please provide a list of the variable names you used from the CMIP output in the section “Attributions of changes in the daily temperature variability”. This is important so that others can easily reproduce your work.

Response: We thank the reviewer for the constructive comments. Following the suggestion of the reviewer, we provide Table S2 showing the variable names you used from the CMIP output in the revised manuscript:

Lines 600-601: The model output variables required to calculate DTD and SDT are listed in Table S2.

Lines 648-649: The model output variables required to calculate TADV and DTDSHF are listed in Table S2.

2.11 L403: Can you provide a reference of the equation used.

Response: Following the suggestion of the reviewer, we add the reference for the equation in the revised manuscript:

Lines 602-605: **Attribution of the change in daily temperature variability.** According to the thermodynamic energy equation⁵⁴, the temperature difference (δT) between two neighboring days can be expressed as:

$$\delta T = \frac{\partial T}{\partial t} = -\mathbf{u} \cdot \nabla T + \left(\frac{RT}{c_p P} - \frac{\partial T}{\partial P} \right) \omega + \frac{1}{c_p} \frac{\partial Q}{\partial t} \quad (3)$$

2.12 Figures: Please add a unit label to the color bar! For Figure 5, also a legend arrow for wind speed is missing. In the current version, the reader cannot assess how strong the changes in wind speed actually are.

Response: Following the suggestion of the reviewer, we add a unit label to the color bar for each figure.

REVIEWER COMMENTS

Reviewer #1 (Remarks to the Author):

From my perspective, the authors have provided a comprehensive and thorough job of responding to my constructive criticism. I can understand the concerns of the other reviewers that perhaps not all of the conclusions are as robust as one might hope due to model output availability and signal to noise concerns. However, I would personally still support publication of this work because of the new focus on the impact of deforestation on variability, which is an interesting potential impact. The authors have correctly revised the paper to be more modest in their conclusions and in their statements of broader impacts.

While the writing is relatively good, this paper could still use some careful editing for meaning, clarity, and grammar.

Reviewer #2 (Remarks to the Author):

General comments:

I'd like to acknowledge the enormous amount of work the authors have put into these revisions. It is clear that they took the advice of each of the reviewers as a guide on how to improve the paper; as a result, this manuscript is substantially improved from the earlier version. The introduction, motivation, and rationale for the focus on temperature variability seems much more complete and the societal importance is clearly articulated. The authors also added many new lines of evidence to support their conclusions about the impact of widespread forest cover change on surface temperature variability. I think there is an enormous amount of valuable, quality work covered in this document, and I commend the authors on these improvements.

However, I still feel the manuscript is not ready for publication, and I am still not convinced that Nature Communications is the right outlet for this work, though I would not protest if this opinion is not shared by the editors or other reviewers.

The key outstanding issue, I feel, is that many of the new lines of evidence (both observational and model simulations) are not adequately described or clearly presented. This may be simply a result of

these sections of the paper being so new: the text is less complete and less clear. There are so many new experiments considered, and they are not clearly delineated or described; it is difficult to know which results (and which models) are contributing to each of the figures. Additionally, both the satellite and FLUXNET dataset sets are not adequately described. Detailed comments are provided below.

Detailed comments:

- In both the Abstract and the Introduction you mention that biogeophysical effects of deforestation on temperature are broadly documented. However, you fail to mention that there is not broad consensus between models about what those effects are, particularly in the mid-latitudes. Papers about the LUCID experiments demonstrated this (e.g., Pitman et al., 2009, 2012; De Noblet-Ducoudre et al. 2012), and Findell et al. (2007) gave a process-based explanation about the relative importance of a model's albedo-driven cooling effect with deforestation compared with its water-availability/limitation warming effect. These papers all pre-dated the important observational advances in some of the papers you do cite, but the fact remains that models have very different mid-latitude mean responses to deforestation. You might want to cite that Findell et al (2007) paper for context about the balance of the albedo and water availability effects accompanying deforestation.

- o Specifically, on line 38 you mention ET efficiency, but it's much more than efficiency: changes in rooting characteristics (especially depth) impact fundamental access to water, and this has enormous implications for the temperature impact of deforestation.

- Given the paper's emphasis on the impact of enhanced near-surface horizontal temperature advection, I think you need to discuss the importance of surface contrasts in driving mesoscale circulations. The length scale of those contrasts (and the resolution of the model!) is important (e.g., Hohenegger et al 2009; Froidevaux et al 2014), as is subgrid heterogeneity and the background wind (e.g., Rochetin et al 2016). This is a difficult issue to address with such large-scale changes, but you need to mention it somewhere. The resolution-dependence of surface impacts on convection discussed in the Hohenegger et al (2009) paper is particularly important to consider and cite.

- When discussing the primary simulations analyzed and the deforestation scenario depicted in Figure 1b, you do not mention the analysis period of the simulations. Is it years 51-80? You also do not mention if dynamic vegetation is turned on in any of the models and if this would allow for forest regrowth. I assume it is turned off, but you should mention this explicitly.

- Figure 1: The blue colors in panel a are too similar for my eye (or printer/screen), especially the two colors for the range -0.3 to -0.5: I can't tell them apart.

- Given the earlier bullet about model differences, discussion of Figure S2 should include mention of model consistency/inconsistency in the signal. Figure S2 and the associated data should be more thoroughly discussed and explained.

- Figure 2 is excellent! The statistical testing and the associated text provide compelling lines of evidence that this topic is worthy of study.
- I still feel that Figures like Figure 3 need to be in equivalent units, since much of your rationale is based on the TADV signal outweighing the DTDSHF signal when these terms are added together. However, I appreciate the authors' attention to this point in the text and in their response to this comment from the earlier review. You should add explicit mention (somewhere around line 629 when you talk about using this for qualitative estimates of term3) that this qualitative approach prevents you from using equivalent units.
- I feel that Figure S5 should not be buried in the supplement. Or it least it needs a much clearer description and discussion in the main text. The paragraph beginning on line 165 is not complete. In particular, upon first read-through I assumed you were still talking about model data – comparing adjacent forested and open tiles within a model grid cell. This was of course difficult to reconcile with the complete deforestation scenario, so I was initially confused. I had to go to the supplement to understand what you were doing. You need to explicitly mention that these are FLUXNET and AmeriFlux field sites so the reader is alerted that you are not just talking about modeling experiments.
- Figure S7: The panel for the model with no tree cover data should not be presented as if the tree cover change is $\pm 5\%$: the reader shouldn't have to go to the caption to learn that the data are not available. Make this clear in the figure panel (e.g., with bold text on top of the empty map).
- In discussing Figure 5, you mention that the ACCESS-only results that are used in panel e are spatially different than the pattern expected from the multi-model result in panel c because the deforestation pattern in ACCESS is different than the other models. The best solution is to show the single-model equivalent of panel c to verify that within each model there should be good spatial correspondence between the two quantities. This could be in the supplement if needed.
- The paragraph beginning on line 345 talks about some DAMIP experiments, but I do not think you show any results from these experiment. Should the first line of the next paragraph be a part of this paragraph? (The sentence ending in "not shown".) You then go on to talk about the results of Figure 5f, which seem to be computed using a different set of experiments, but it is not immediately clear to me what runs are used for Figure 5f. On lines 374-5 you talk about 8 models for panels c and d, and 4 models for panel f (and the 1 model for panel e, which you do clearly state). This whole description of Figure 5 needs to be much clearer.
- I also can't differentiate some of the blues in panel a of Figure 5 or the two greenish lines in panel b. (Sorry if my printer is poor quality, but I certainly won't be the only one with this issue.)
- Line 396: You say winter, spring and summer; do you mean winter, spring and autumn?
- Line 645: Typo: cloud instead of could.

Reviewer #3 (Remarks to the Author):

Review of "Deforestation intensifies daily temperature variability in the northern extratropics" by Ge et al.

First, I'd like to thank the authors for addressing my general comments and all my specific comments in a thorough and very comprehensive way. The revised manuscript improved considerably, especially due to the inclusion of the section "Investigation of the biogeophysical effect of historical deforestation" (Response to my General Comment 1.1). Overall the study at hand now provides very comprehensive analyses of many Earth system models, various and new MIPs (ScenarioMIP, LUMIP, etc.), as well as observational datasets. Thus this manuscript constitutes an important piece of work in the ongoing research field of deforestation/afforestation and its effects in the Earth system via land-atmosphere coupling. I am still not fully convinced, that the authors could show that the effect of deforestation on daily temperature variability is significant and strong enough, given the uncertainties, to have "implications for the northern extratropical countries that plan to implement large-scale afforestation" (LL31-32). Also, a few important lines of inquiry remain insufficiently addressed (see my comments). I have a few more comments (see below) with respect to the revised manuscript. If these can be addressed adequately, I can recommend the study for publication in Nature Communications.

General Comments:

1. You write that you investigate the biogeophysical effects of deforestation on daily temperature variability at the global scale in models and in observations. Using the thermodynamic energy equation you state that changes in sensible heat flux and surface advection of warm air are causing the increase in daily temperature variability. Why do you not further look into the mechanisms and explain why DTDSHF and TADV are changing? For the latter, you probably cannot do much without having access to the model and doing some idealized runs yourself, but for the former you should be able to dig deeper. Why is DTDSH decreasing so strongly in spring in the extratropics? Is it because of changes in albedo (changes in snow cover due to deforestation) or changes in the bowen ratio/evaporative fraction (changes in latent heat due to reduced transpiration rooted in the removal of forest)? I think it'd be useful to show changes in the variability in the surface energy balance to see where the changes in DTDSH are coming from.

Specific Comments:

L20: Maybe change to "effects of deforestation on average temperature"?

LL354-355: "At the global scale, the deforestation effects on daily temperature variability are much smaller than the effects of other anthropogenic forcings (not shown)." Thanks for this addition and response to my comment. If you already quantified other anthropogenic drivers, you should compare the estimates quantitatively. So, please show the estimates of each effect to get a better intuition what "much smaller" means.

L157: „popular index“ reads a little strangely. What about “common index” or “more commonly used index”?

LL528-529: “Fortunately, we find a large area of forest losses during the historical period in North America (Fig. 5a).” Maybe “fortunately” is the wrong word here (I hope you don’t really find it fortunate to find large losses of forests...). I know what you mean though. Maybe better “However, we could identify a large area...”.

L410: This section reads more like “Discussion & Conclusions”.

Response to reviewer

We would like to thank the three reviewers for their insightful and constructive comments. We believe that we have satisfactorily addressed all of the major and minor comments raised by the reviewers. Our point-by-point responses to the comments are listed below in blue.

Reviewer #1 (Remarks to the Author):

From my perspective, the authors have provided a comprehensive and thorough job of responding to my constructive criticism. I can understand the concerns of the other reviewers that perhaps not all of the conclusions are as robust as one might hope due to model output availability and signal to noise concerns. However, I would personally still support publication of this work because of the new focus on the impact of deforestation on variability, which is an interesting potential impact. The authors have correctly revised the paper to be more modest in their conclusions and in their statements of broader impacts.

Response: Thank you for the constructive comments in the last round of review, which help to substantially improve the manuscript.

While the writing is relatively good, this paper could still use some careful editing for meaning, clarity, and grammar.

Response: We further polish the manuscript in terms of meaning, clarity and grammar with help from the Nature Research Editing Service.

Reviewer #2 (Remarks to the Author):

General comments:

I'd like to acknowledge the enormous amount of work the authors have put into these revisions. It is clear that they took the advice of each of the reviewers as a guide on how to improve the paper; as a result, this manuscript is substantially improved from the earlier version. The introduction, motivation, and rationale for the focus on temperature variability seems much more complete and the societal importance is clearly articulated. The authors also added many new lines of evidence to support their conclusions about the impact of widespread forest cover change on surface temperature variability. I think there is an enormous amount of valuable, quality work covered in this document, and I commend the authors on these improvements.

Response: Thank you for the constructive comments in the last round of review, which help to substantially improve the manuscript.

However, I still feel the manuscript is not ready for publication, and I am still not convinced that Nature Communications is the right outlet for this work, though I would not protest if this opinion is not shared by the editors or other reviewers.

The key outstanding issue, I feel, is that many of the new lines of evidence (both observational and model simulations) are not adequately described or clearly presented. This may be simply a result of these sections of the paper being so new: the text is less complete and less clear. There are so many new experiments considered, and they are not clearly delineated or described; it is difficult to know which results (and which models) are contributing to each of the figures. Additionally, both the satellite and FLUXNET dataset sets are not adequately described. Detailed comments are provided below.

Detailed comments:

1. In both the Abstract and the Introduction you mention that biogeophysical effects of deforestation on temperature are broadly documented. However, you fail to mention that there is not broad consensus between models about what those effects are, particularly in the mid-latitudes. Papers about the LUCID experiments demonstrated this (e.g., Pitman et al., 2009, 2012; De Noblet-Ducoudre et al. 2012), and Findell et al. (2007) gave a process-based explanation about the relative importance of a model's albedo-driven cooling effect with deforestation compared with its water-availability/limitation warming effect. These papers all pre-dated the important observational advances in some of the papers you do cite, but the fact remains that models have very different mid-latitude mean responses to deforestation. You might want to cite that Findell et al (2007) paper for context about the balance of the albedo and water availability effects accompanying deforestation. Specifically, on line 38 you mention ET efficiency, but it's much more than efficiency: changes in rooting characteristics (especially depth) impact fundamental access to water, and this has enormous implications for the temperature impact of deforestation.

Response: Following the suggestion of this reviewer, we have added more information on the model uncertainty in the biogeophysical effect and the mechanisms for the biogeophysical effects to the Introduction section:

Line 39-50: In addition to the impact on the carbon cycle, deforestation also has profound impacts on local and regional climate through biogeophysical processes³. On the one hand, deforestation enlarges the surface albedo and cools the climate; on the other hand, deforestation diminishes evapotranspiration and warms the climate due to the lower aerodynamic roughness, rooting depth, leaf area and canopy conductance for transpiration⁴⁻⁶. The albedo-driven cooling effect dominates boreal regions, whereas the evapotranspiration-driven warming effect dominates the tropics⁷⁻¹⁰. In the mid-latitudes, however, the deforestation effect is complicated and uncertain because of the lower model agreement in this regard¹¹⁻¹².

Accordingly, we have also added the citation of the three important papers (Pitman et al. 2009; de Noblet-Ducoudre et al. 2012; Findell et al. 2007) recommended by the reviewer into the revised manuscript.

2. Given the paper's emphasis on the impact of enhanced near-surface horizontal temperature advection, I think you need to discuss the importance of surface contrasts in driving mesoscale circulations. The length scale of those contrasts (and the resolution of the model!) is important (e.g., Hohenegger et al 2009; Froidevaux et al 2014), as is subgrid heterogeneity and the background wind (e.g., Rochetin et al 2016). This is a difficult issue to address with such large-scale changes, but you need to mention it somewhere. The resolution-dependence of surface impacts on convection discussed in the Hohenegger et al (2009) paper is particularly important to consider and cite.

Response: Thank you for this constructive comment. Following the suggestion of this reviewer, we have added a discussion of the uncertainty related to the length scale of surface contrasts and model resolution to the revised manuscript:

Line 548-565: We identify the enhanced near-surface horizontal wind speed and temperature advection as the main drivers of the higher daily temperature variability over deforested areas. It should be emphasized that the response of wind to deforestation may depend on the deforestation scale. In this study, the enhanced wind speed and temperature advection should be regarded as the consequence of large-scale deforestation. However, small-scale deforestation favors the generation of mesoscale circulations due to the higher sensible heat flux^{60, 61}. Such thermodynamically driven mesoscale circulations may further regulate the wind and temperature advection over deforested areas. Moreover, the model resolution has also been proven to play an important role in determining the simulated response of mesoscale circulations to surface perturbations^{62, 63}. In particular, the high-resolution (< 4 km) simulation with an explicit treatment of convection and the coarse-resolution (~10 to ~100 km) simulation with parameterized convection can even produce totally different responses of

mesoscale circulations to the same surface perturbation⁶². In this study, only coarse-resolution earth system models are used, and the results are probability model resolution dependent. Therefore, high-resolution simulations (e.g., the convection-permitting simulation) are encouraged to further investigate the deforestation effect on daily temperature variability, particularly the effect of small-scale deforestation.

Accordingly, we have also added the citation of the important paper (Hohenegger et al. 2009) recommended by the reviewer into the revised manuscript.

3. When discussing the primary simulations analyzed and the deforestation scenario depicted in Figure 1b, you do not mention the analysis period of the simulations. Is it years 51-80? You also do not mention if dynamic vegetation is turned on in any of the models and if this would allow for forest regrowth. I assume it is turned off, but you should mention this explicitly.

Response: Following the suggestion of the reviewer, we specifically mention the analysis period of the simulations and the configuration of dynamic vegetation in the revised manuscript.

Line 120-125: Vegetation dynamics are required to be switched off in the deforested grid cells for *deforest-globe*⁴⁵. The biogeophysical effect of the idealized deforestation can be isolated through the comparison of the last 30-year (years 51 to 80) simulations of *piControl* and *deforest-globe* (*deforest-globe* minus *piControl*; see “Methods”).

4. Figure 1: The blue colors in panel a are too similar for my eye (or printer/screen), especially the two colors for the range -0.3 to -0.5: I can't tell them apart.

Response: We apologize for this inconvenience. We have updated the color scheme for Figure 1 in the revised manuscript. We hope that the new color scheme can make the figure reader-friendlier.

5. Given the earlier bullet about model differences, discussion of Figure S2 should

include mention of model consistency/inconsistency in the signal. Figure S2 and the associated data should be more thoroughly discussed and explained.

Response: Following the suggestion of this reviewer, we have added the mention of model consistency/inconsistency in the signal shown in Figure S2 to the revised manuscript.

Line 150-153: The models consistently suggest a cooling effect of deforestation in the northern extratropics (Supplementary Figs. 2a-2d). In the tropics, however, the models show less agreement on the deforestation effect, and the multimodel mean suggests a warming effect of deforestation.

We have also added a thorough discussion and explanation of Figure S2 and the associated data to the revised manuscript.

Line 145-150: We also evaluate the model representation of the biogeophysical effect of deforestation on mean surface temperature with a satellite-based dataset^{13, 48}. This dataset provides the observed biogeophysical effect of deforestation at the global scale based on the “space-for-time” substitution method, that is, the comparison of the spatially adjacent forest and openland pixels⁴⁸ (see “Methods”).

Line 156-165: Note that, in contrast to the observations, the models simulate a more widespread cooling effect of deforestation in the northern mid-latitudes. This discrepancy is mainly attributed to the fact that the deforestation-induced atmospheric feedbacks (e.g., cloud formation) to surface temperature are fully considered in the models but absent in the observed signal due to the application of the “space-for-time” method^{49, 50}; this method assumes identical background climate over the paired forest and openland pixels. It has been demonstrated that the atmospheric feedbacks are considerable and mostly explain the simulated cooling effect of deforestation in the northern mid-latitudes^{49, 50}.

6. Figure 2 is excellent! The statistical testing and the associated text provide compelling lines of evidence that this topic is worthy of study.

Response: Thank you!

7. I still feel that Figures like Figure 3 need to be in equivalent units, since much of your rationale is based on the $\Delta TADV$ signal outweighing the $\Delta DTDSHF$ signal when these terms are added together. However, I appreciate the authors' attention to this point in the text and in their response to this comment from the earlier review. You should add explicit mention (somewhere around line 629 when you talk about using this for qualitative estimates of $\Delta term3$) that this qualitative approach prevents you from using equivalent units.

Response: Thank you for your understanding and suggestion. We have added an explicit mention to the revised manuscript.

Line 758-760: It should be emphasized that this approach prevents us from showing $\Delta term3$ in a unit equivalent to ΔTDT .

8. I feel that Figure S5 should not be buried in the supplement. Or it least it needs a much clearer description and discussion in the main text. The paragraph beginning on line 165 is not complete. In particular, upon first read-through I assumed you were still talking about model data – comparing adjacent forested and open tiles within a model grid cell. This was of course difficult to reconcile with the complete deforestation scenario, so I was initially confused. I had to go to the supplement to understand what you were doing. You need to explicitly mention that these are FLUXNET and AmeriFlux field sites so the reader is alerted that you are not just talking about modeling experiments.

Response: We have moved some important information in original Fig. S5 to Figs. 1, 3 and 4 in the main text in the revised manuscript. Please see the box plots imbedded in the panels of Figs. 1, 3 and 4.

We have also added the explicit mention of the FLUXNET and AmeriFlux field sites to this paragraph in the revised manuscript.

Line 203-209: In addition to the simulations, we attempt to find observational evidence on the deforestation effect on daily temperature variability. To this end, 24 selected pairs of spatially adjacent forest and openland sites in North America and Europe (Supplementary Fig. 5; Supplementary Table 3) from the FLUXNET and AmeriFlux datasets are used⁵¹⁻⁵³ (see “Methods”). The box plots imbedded in Figs. 1c-1f show the DTDT differences between the paired forest and openland sites (openland minus forest) for each season.

9. Figure S7: The panel for the model with no tree cover data should not be presented as if the tree cover change is $\pm 5\%$: the reader shouldn't have to go to the caption to learn that the data are not available. Make this clear in the figure panel (e.g., with bold text on top of the empty map).

Response: Following the suggestion of this reviewer, we have added the bold text “Data not available” to the empty map. Please see Supplementary Figure 7b.

10. In discussing Figure 5, you mention that the ACCESS-only results that are used in panel e are spatially different than the pattern expected from the multi-model result in panel c because the deforestation pattern in ACCESS is different than the other models. The best solution is to show the single-model equivalent of panel c to verify that within each model there should be good spatial correspondence between the two quantities. This could be in the supplement if needed.

Response: Following the suggestion of this reviewer, we have added a figure showing the DTDT change from the ACCESS model to the supplementary material. Please see Supplementary Fig. 8. Accordingly, we have also rewritten this part.

Line 429-435: Note that there is some spatial mismatch between the DTDT change (Fig. 5b) and the TADV change (Fig. 5d) in North America. This mismatch is because the

pattern of tree cover changes for the ACCESS-ESM1-5 model (Supplementary Fig. 7a) is slightly different from the multimodel mean pattern (Fig. 5a). Examining the ACCESS-ESM1-5 model, we find that the DTD and TADV changes from this model overall match each other in terms of the geographical pattern (Figs. 5d and Supplementary Fig. 8).

11. The paragraph beginning on line 345 talks about some DAMIP experiments, but I do not think you show any results from these experiment. Should the first line of the next paragraph be a part of this paragraph? (The sentence ending in “not shown”.) You then go on to talk about the results of Figure 5f, which seem to be computed using a different set of experiments, but it is not immediately clear to me what runs are used for Figure 5f. On lines 374-5 you talk about 8 models for panels c and d, and 4 models for panel f (and the 1 model for panel e, which you do clearly state). This whole description of Figure 5 needs to be much clearer.

Response: We agree, and we combine these two paragraphs (Line 436-458). We have also rewritten the description of Fig. 5 to clarify the number of models that are used to plot each panel.

Line 468-479: The DTD and DTDTSHF changes are from the mean value of eight models, and the black dots in panels b and c indicate that at least seven out of the eight models agree on the sign of the change. The TADV change is from the ACCESS-ESM1-5 model, and the black dots in panel d indicate that the TADV change is statistically significant at the 95% confidence level tested by two-tailed Student’s *t* test. e The net effects of all anthropogenic forcings (GHGs, aerosols and LULCCs) on wintertime DTD during the historical period. In panel e, the DTD change is from the mean value of four models, and the black dots indicate that all four models agree on the sign of the change.

12. I also can’t differentiate some of the blues in panel a of Figure 5 or the two greenish lines in panel b. (Sorry if my printer is poor quality, but I certainly won’t be the

only one with this issue.)

Response: This is also a comment related to the color scheme. We have updated the color scheme for all figures in the revised manuscript to make these figures reader-friendlier.

13. Line 396: You say winter, spring and summer; do you mean winter, spring and autumn?

Response: We apologize for the typo. We have changed “summer” to “autumn” (Line 500).

14. Line 645: Typo: cloud instead of could.

Response: We have changed “could” to “cloud” (Line 773).

Reviewer #3 (Remarks to the Author):

Review of "Deforestation intensifies daily temperature variability in the northern extratropics" by Ge et al.

First, I'd like to thank the authors for addressing my general comments and all my specific comments in a thorough and very comprehensive way. The revised manuscript improved considerably, especially due to the inclusion of the section "Investigation of the biogeophysical effect of historical deforestation" (Response to my General Comment 1.1). Overall the study at hand now provides very comprehensive analyses of many Earth system models, various and new MIPs (ScenarioMIP, LUMIP, etc.), as well as observational datasets. Thus this manuscript constitutes an important piece of work in the ongoing research field of deforestation/afforestation and its effects in the Earth system via land-atmosphere coupling. I am still not fully convinced, that the authors could show that the effect of deforestation on daily temperature variability is significant and strong enough, given the uncertainties, to have "implications for the northern extratropical countries that plan to implement large-scale afforestation" (LL31-32). Also, a few important lines of inquiry remain insufficiently addressed (see my comments). I have a few more comments (see below) with respect to the revised manuscript. If these can be addressed adequately, I can recommend the study for publication in Nature Communications.

Response: Thank you for the constructive comments in the last round of review, which help to substantially improve the manuscript.

General Comments:

1. You write that you investigate the biogeophysical effects of deforestation on daily temperature variability at the global scale in models and in observations. Using the thermodynamic energy equation you state that changes in sensible heat flux and surface advection of warm air are causing the increase in daily temperature

variability. Why do you not further look into the mechanisms and explain why DTDSHF and TADV are changing? For the latter, you probably cannot do much without having access to the model and doing some idealized runs yourself, but for the former you should be able to dig deeper. Why is DTDSH decreasing so strongly in spring in the extratropics? Is it because of changes in albedo (changes in snow cover due to deforestation) or changes in the bowen ratio/evaporative fraction (changes in latent heat due to reduced transpiration rooted in the removal of forest)? I think it'd be useful to show changes in the variability in the surface energy balance to see where the changes in DTDSH are coming from.

Response: Thank you for this constructive comment. Following the suggestion of this reviewer, we further analyze the variabilities in the components of the surface energy balance. We find that the higher springtime DTDSHF change is mainly related to the albedo effect. Please see Fig. 4 and the revised text:

Line 317-331: To explain the seasonal variation in the DTDSHF response to deforestation, we further examine the daily variabilities in the other components (including shortwave and longwave radiations and latent heat flux) of the surface energy balance based on the CanESM model given data availability. As shown in Figs. 4e-4l, the daily variabilities in surface upward shortwave radiation (DTDUSR) and albedo (DTD α) increase in deforested areas. This result indicates that deforestation enlarges not only the surface albedo and the reflected solar radiation, but also their variabilities. In the northern extratropic deforested areas, the DTDUSR change is larger in spring because of the larger DTD α change due to the snow cover as well as the higher springtime solar radiation. The larger DTDUSR change mostly explains the larger DTDSHF change in the northern extratropic deforested areas in spring. Moreover, the simulated responses of DTDUSR and DTD α to deforestation are also supported by the paired site observations (see the box plots embedded in Figs. 4e-4l). The daily variabilities in surface net longwave radiation and latent heat flux are less influenced by deforestation (not shown).

Specific Comments:

1. L20: Maybe change to “effects of deforestation on average temperature”?

Response: Following the suggestion of this reviewer, we have changed “effects of deforestation on temperature” to “effects of deforestation on average and extreme temperatures” (Lines 23-24).

2. LL354-355: “At the global scale, the deforestation effects on daily temperature variability are much smaller than the effects of other anthropogenic forcings (not shown).” Thanks for this addition and response to my comment. If you already quantified other anthropogenic drivers, you should compare the estimates quantitatively. So, please show the estimates of each effect to get a better intuition what “much smaller” means.

Response: Thank you for this constructive comment. We have added a figure showing the quantitative contributions of anthropogenic forcings to the revised manuscript. Please see Fig. 5f and the revised text:

Line 444-447: Averaged over the global land, the deforestation effect on DTDT (0.01 °C) is much smaller than the combined effect of GHGs and aerosols (-0.03 °C) in magnitude (Fig. 5f).

Line 456-458: Averaged over the deforested area of North America, the contributions of deforestation and the combined GHG and aerosol forcings to the historical DTDT change are 0.11 and -0.17 °C, respectively (Fig. 5f).

3. L157: „popular index“ reads a little strangely. What about “common index” or “more commonly used index”?

Response: Following the suggestion of this reviewer, we have changed “popular index” to “more commonly used index” (Line 195).

4. LL528-529: “Fortunately, we find a large area of forest losses during the historical period in North America (Fig. 5a).” Maybe “fortunately” is the wrong word here (I hope you don’t really find it fortunate to find large losses of forests...). I know what you mean though. Maybe better “However, we could identify a large area...”.

Response: Following the suggestion of this reviewer, we have changed “Fortunately, we find ...” to “However, we identify a large area ...” (Lines 652-653).

5. L410: This section reads more like “Discussion & Conclusions”.

Response: Following the suggestion of this reviewer, we have changed the title of this section to “Discussion & Conclusions” (Line 515).

REVIEWERS' COMMENTS

Reviewer #2 (Remarks to the Author):

Most of this paper is very strong now. It is clear from the new manuscript and from the responses to the reviews that the authors have worked hard to bring the document up to the standards of Nature Communications. The introductory section is excellent; the historical and future scenario sections are very nicely done; the discussion and conclusions are clear and strong.

However, the comparison with Fluxnet sites is still concerning. I don't find the explanation about $\Delta TADV$ and DTDSHF (lines 332-340) to be satisfactory. The rationale in that paragraph is difficult to understand, particularly for boreal autumn.

- On lines 336-337 you say, "the DTDSHF response to deforestation is larger in spring, summer and autumn (Figs. 4a-4d)." Larger than what? Larger than the DTDSHF signal in winter? That doesn't appear to be true (4d vs 4a). I think you mean larger than the $\Delta TADV$ signal, but it is not clear.
- Additionally, on lines 334-336 you say, "the atmosphere above the paired forest and openland sites tends to be mixed due to horizontal advection, alleviating the $\Delta TADV$ difference between the paired sites." Isn't this true for all seasons? This doesn't make sense as an explanation for behavior in 3 seasons but not 4.

In the scope of the full paper, this critique seems minor. I applaud the authors for making such great strides. I think this one section should be improved, and then I would support publication in Nature Communications.

Detailed suggestions:

- Figure 1 caption (and elsewhere): season names only work for one hemisphere. Use DJF, MAM, etc.
- Line 508: "probability" is probably meant to be "probably".

Reviewer #3 (Remarks to the Author):

I again thank the authors for addressing all my comments and modifying the manuscript accordingly. I would like to ask the authors to publish their code and processed model output on public servers, such as <https://zenodo.org/> . Other than that, I have no further comments and recommend the manuscript for publication.

Alexander J Winkler

Response to reviewers

We would like to thank the two reviewers for their encouraging and constructive comments. We have satisfactorily addressed the minor comments raised by the two reviewers. Our point-by-point responses to the reviewers' comments are listed below in blue.

Reviewer #2 (Remarks to the Author):

Most of this paper is very strong now. It is clear from the new manuscript and from the responses to the reviews that the authors have worked hard to bring the document up to the standards of Nature Communications. The introductory section is excellent; the historical and future scenario sections are very nicely done; the discussion and conclusions are clear and strong.

Response: Thank you for the constructive comments in the last two rounds of reviews, which helped us to substantially improve the manuscript.

However, the comparison with Fluxnet sites is still concerning. I don't find the explanation about $\Delta(TADV)$ and DTDSHF (lines 332-340) to be satisfactory. The rationale in that paragraph is difficult to understand, particularly for boreal autumn.

- On lines 336-337 you say, "the DTDSHF response to deforestation is larger in spring, summer and autumn (Figs. 4a-4d)." Larger than what? Larger than the DTDSHF signal in winter? That doesn't appear to be true (4d vs 4a). I think you mean larger than the $\Delta(TADV)$ signal, but it is not clear.

Response: We mean larger than the DTDSHF signal in winter. We agree with the reviewer that the signal in autumn is not obviously larger than that in winter from the in situ observation. Therefore, we rewrote this sentence.

Line 339-342: Thus, the DTDT differences between the paired sites are more influenced by the DTDSHF effect, particularly in spring and summer when the TADV effect is smaller (inferred from the simulations) but the DTDSHF effect is larger (Figs. 4a-4d).

- Additionally, on lines 334-336 you say, "the atmosphere above the paired forest and openland sites tends to be mixed due to horizontal advection, alleviating the TADV difference between the paired sites." Isn't this true for all seasons? This doesn't make sense as an explanation for behavior in 3 seasons but not 4.

Response: Theoretically, this is true for all seasons, and we emphasize this in the revised manuscript:

Line 341-343: Specifically, the atmosphere above the paired forest and openland sites tends to be mixed due to horizontal advection, alleviating the TADV difference between the paired sites throughout the whole year.

We have also slightly reorganized this paragraph to make it clearer to readers.

In the scope of the full paper, this critique seems minor. I applaud the authors for making such great strides. I think this one section should be improved, and then I would support publication in Nature Communications.

Response: Thank you. We have further polished this section based on the comments listed above.

Detailed suggestions:

- Figure 1 caption (and elsewhere): season names only work for one hemisphere. Use DJF, MAM, etc.

Response: Following the suggestion of this reviewer, we have replaced season names in Figure captions with DJF, MAM, etc. For example:

Line 170-172: in (c) DJF (December, January and February), (d) MAM (March, April and May), (e) JJA (June, July and August) and (f) SON (September, October and November).

- Line 508: “probability” is probably meant to be “probably”.

Response: The typo has been corrected (Line 512).

Reviewer #3 (Remarks to the Author):

I again thank the authors for addressing all my comments and modifying the manuscript accordingly. I would like to ask the authors to publish their code and processed model output on public servers, such as <https://zenodo.org/> . Other than that, I have no further comments and recommend the manuscript for publication.

Response: Thank you for the constructive comments in the last two rounds of review, which helped us to substantially improve the manuscript.

Following the policies of Nature Communications, we have uploaded our code and the processed model output to a public repository.

Alexander J Winkler